# JANUS: DUAL-SERVER MULTI-ROUND SECURE AGGREGATION WITH VERIFIABILITY FOR FEDERATED LEARNING

## ABSTRACT

Secure Aggregation (SA) in federated learning is essential for preserving user privacy by ensuring that model updates are masked or encrypted and remain inaccessible to servers. Although the advanced protocol Flamingo (S&P'23) has made significant strides with its multi-round aggregation and optimized communication, it still faces several critical challenges: (i) *Dynamic User Participation*, where Flamingo struggles with scalability due to the complex setups required when users join or leave the training process; (ii) *Model Inconsistency Attacks* (MIA), where a malicious server could infer sensitive data, which poses severe privacy risks; and (iii) *Verifiability*, as most schemes lack an efficient mechanism for clients to verify the correctness of server-side aggregation, potentially allowing inaccuracies or malicious actions. We introduce Janus, a generic privacy-enhanced multi-round SA scheme through a dual-server architecture. A new user can participate in training by simply obtaining the servers' public keys for aggregation, eliminating the need for complex communication graphs. Our dual-server model separates aggregation tasks, ensuring that neither server can successfully launch a MIA without controlling at least $n-1$ clients. Additionally, we propose a new cryptographic primitive, *Separable Homomorphic Commitment*, integrated with our dual-server approach to ensure the verifiability of aggregation results. Extensive experiments across various models and datasets show that Janus significantly boosts security while enhancing efficiency. It reduces per-client communication and computation overhead from logarithmic to constant scale compared to state-of-the-art methods, with almost no compromise in model accuracy.

## 1 INTRODUCTION

Traditional machine learning relies on centralized training, where the entire dataset is stored in a single central location and directly accessible by the server. However, users are often reluctant to share data, especially if it involves sensitive information like medical records, photos, or trade secrets. Federated Learning (FL) was proposed to protect user privacy and enable the model training (McMahan et al., 2017). FL is a distributed machine learning framework that uses privacy-preserving cryptographic techniques, which allows participants to collaborate on model training without disclosing their private data. Unfortunately, it has been shown that an adversary can invert a single model update from a target user, thereby revealing a great deal of sensitive information about its local dataset (Hitaj et al., 2017; Nasr et al., 2019; Zhu et al., 2019).

To protect the user gradient information, Secure Aggregation (SA (Bonawitz et al., 2017)) is introduced to enhance the security of FL, which can prevent server access to individual model updates. SA is considered as one of the most robust defenses against gradient inversion and related inference attacks (Huang et al., 2021). Most of the current SA schemes rely on the double-mask, which involves heavy secret sharing, especially as the number of participants grows, requiring two clients to negotiate the key and engage in frequent communication. The advanced SA protocol (BBSA (Bell et al., 2020)) manages the aggregation with thousands of clients and high-dimensional input vectors while tolerating device drops during execution. However, these schemes select a subset of clients and enables aggregation for only one round. Although it is possible to run the protocol multiple times to complete multi-round of aggregation, the *Setup* phase must be re-run for each round to

maintain privacy, requiring server interaction with all clients during each step. This results in significant communication overhead and reduced efficiency.

Recently, the state-of-the-art Flamingo (Ma et al., 2023) eliminates the need for re-setup in each round, which supports multi-round SA based on the BBSA. It also optimizes the communication graph to improve the system performance, with introducing a set of decryptors to handle part of the computation. While Flamingo marks significant progress, it has limitations in handling dynamic user participation, resisting Model Inconsistency Attacks (MIA) (Pasquini et al., 2022), and ensuring correct server-side aggregation. When users join or leave, the complex setups needed for Flamingo lessen its practicality. The server can still exploit the MIA to infer sensitive information, and clients have no way to verify if the server correctly performed the aggregation or omitted user data.

These vulnerabilities stem from the reliance on a single server, which is common in existing schemes due to its simplicity. A single server inherently knows the aggregated results, providing an opportunity for a malicious server to compromise the privacy by bypassing the SA protocol (Pasquini et al., 2022). Specifically, the server distributes carefully crafted parameters to non-target users, which can trigger the *dying-ReLU* effect that causes non-target users to generate zero gradients during aggregation. As a result, the aggregated gradient effectively reveals the target user's gradient. This attack affects not only double-mask schemes but all schemes where the server can access the aggregation results. While cryptographic signatures could prevent this by allowing users to verify the consistency of received parameters. This approach involves heavy computation and requires users to negotiate the consistency of the received information, which places a large burden on the system.

Our research indicates that preventing MIAs necessitates restricting the server's access to the final aggregation results. To achieve this, we propose a dual-server architecture: one server handles the collection and aggregation of masked gradients, while the other manages the aggregation of masks. If the servers do not collude, neither can access the final aggregated results. This assumption is feasible in many real-world scenarios. For example, banks, financial institutions, and healthcare organizations, despite having different interests, are generally committed to protecting user privacy and complying with regulations. They are motivated to collaborate for the benefit of users and avoid collusion. In the Flamingo scheme, the decryptors can be also considered as one server, with a second server forming the dual-server architecture. This approach ensures security while leveraging the practical willingness of institutions to cooperate for SA. Additionally, numerous studies are relevant to our work, with detailed discussion provided in Appendix A.

Another challenge is to ensure the correctness of aggregation, particularly in a dual-server architecture where either server could miscollect or misaggregate masked gradients or masks. An aggregation server might prioritize speed over accuracy, performing fast but faulty computations to save resources, which can lead to erroneous results. Since servers are often semi-trusted, they could also deliberately mishandle some gradients or falsify aggregation results, misleading users about the training results (Hahn et al., 2023). Current schemes face difficulties in ensuring efficient verifiability, typically depending on resource-intensive techniques like homomorphic hashing or signatures. Moreover, errors in aggregation could arise from malicious client submissions, yet current methods fail to enforce strong client-side commitments. To address these challenges, our approach introduces a new cryptographic primitive called separable homomorphic commitment (SHC), which ensures both server-side integrity and client-side data accuracy in the dual-server setting. Homomorphism and separability are two important properties of SHC. The two servers aggregate the different values in the commitment separately. SHC can separates out the part of message and compares them with the aggregated results, thus enabling the correctness of aggregation.

Our main contributions are summarized as follows.

- *Generic construction of dual-server SA with dynamic user participation for FL.* We propose Janus, the first generic construction of SA based on dual-server, which can work well for multiple round of aggregation without re-setup in FL. Our new design avoids heavy communication graphs such as complete graphs and $k$-regular graphs. Additionally, Janus only involves some lightweight components, thus it can avoid the need for time-consuming operations such as secret sharing, which in turn dramatically improves the system efficiency. It also enables dynamic user participation with only the servers' public keys.
- *A new cryptographic primitive and enhanced privacy with verifiability.* Our primary contribution is the conceptual development of a new cryptographic primitive, termed *Separable*

*Homomorphic Commitment* (SHC). By analyzing the algebraic properties of current commitment schemes, we identify a common blueprint that can be instantiated to provide novel verification methods for aggregation results. Furthermore, we introduce a dual-server architecture that leverages SHC to enhance both privacy and verifiability. This architecture ensures that aggregation results remain invisible to individual servers, making it impossible for a malicious server to bypass the SA. Consequently, our approach not only enhances resistance to malicious inference attacks but also incorporates verifiability, providing additional security advantages.

- *Implementation and evaluation.* We implemented an instantiation for Janus and evaluated it with similar classical schemes via extensive experiments on different models and datasets. The results show that Janus outperforms in terms of both computation and communication. It reduces per-client overhead from the logarithmic scale of current advanced methods to a constant scale. Table 1 demonstrates that Janus surpasses other state-of-the-art schemes in terms of security, efficiency, and functionality.

Table 1: Comparison of SA Constructions

| Scheme | Input Privacy | Multi-round | Verifiability | Dynamic | Versatility | NS[*] | Efficiency[‡] | MIA |
|---|---|---|---|---|---|---|---|---|
| SecAgg (Bonawitz et al., 2017) | ✓ | ✗ | ✗ | ✗ | ✗ | 1 | ◯ | ✗ |
| BBSA (Bell et al., 2020) | ✓ | ✗ | ✗ | ✗ | ✗ | 1 | ◐ | ✗ |
| Flamingo (Ma et al., 2023) | ✓ | ✓ | ✗ | ✗ | ✗ | 2[†] | ⬤ | ✗ |
| Janus | ✓ | ✓ | ✓ | ✓ | ✓ | 2 | ⬤ | ✓ |

✓ Support, ✗ No support. Versatility: A generic construction. ⋆ Number of servers. † The decryptors of this construction can be abstracted to a server. ‡ More black parts in the circle indicate better efficiency.

## 2 PRELIMINARIES

### 2.1 COMMITMENTS

Commitments (Pedersen, 1991) provide the cryptographic cornerstone for integrity and trust in various schemes. It enables participants to commit to values without compromising the confidentiality of the information. Typically, a non-interactive secure commitment scheme consists of the following three algorithms:

1. $\mathsf{CSetup}(1^\lambda) \to pp$. The system initialization algorithm takes as input a security parameter $\lambda$, and it outputs the public parameter $pp$ for the commitment scheme.
2. $\mathsf{Commit}(pp, v, r) \to c$. The commitment generation algorithm takes as input a message $v$ from the message space $\mathcal{M}_{pp}$ and a random number (blinder) $r$ in the randomness space $\mathcal{R}_{pp}$, and it outputs the commitment $c$ in the commitment space $\mathcal{C}_{pp}$.
3. $\mathsf{Reveal}(pp, v, c, r) \to b$. The revealing commitment algorithm takes as input a message $v$, a commitment $c$ and a blinder $r$. If it accepts then the output $b = 1$; otherwise, $b = 0$.

Normally, a secure commitment scheme must satisfy the following three properties.

- **Completeness.** It ensures that if both the committer and the verifier follow the protocol correctly, the verifier will always accept the decommitment ($\mathsf{Reveal}$).

$$\Pr \left( \begin{array}{c} \mathsf{CSetup}(1^\lambda) \to pp; \\ \mathsf{Commit}(pp, v, r) \to c: \\ \mathsf{Reveal}(pp, v, c, r) = 1 \end{array} \right) = 1. \quad (1)$$

- **Hiding.** During the commitment phase, the verifier cannot infer the committed value from the commitment. It can ensure that the committed value remains confidential until it is revealed. For any $v_1, v_2$ of equal length, and any $r$, the following probability distributions are computationally indistinguishable.

$$\{\mathsf{Commit}(pp, v_1, r) \to c_1\} \stackrel{c}{\approx} \{\mathsf{Commit}(pp, v_2, r) \to c_2\}. \quad (2)$$

- **Binding.** After the commitment is made, the committer cannot change the committed value. It can prevent the committer from cheating by ensuring the immutability of the commitment. There exists a negligible function $\mathsf{negl}(\lambda)$ such that for all non-uniform Probabilistic Polynomial Time ($\mathcal{PPT}$) adversaries $\mathcal{A}$,

$$\Pr \left( \begin{array}{c} \mathsf{CSetup}(1^\lambda) \to pp; \\ \mathcal{A}(pp) \to (c, r, v_1, v_2): \\ \mathsf{Reveal}(pp, c, v_1, r) = 1 \wedge \\ \mathsf{Reveal}(pp, c, v_2, r) = 1 \wedge \\ v_1 \neq v_2 \end{array} \right) \leq \mathsf{negl}(\lambda). \tag{3}$$

## 2.2 MASKING-BASED SECURE AGGREGATION

The One-Time Pad (OTP) is a type of classical encryption which can be perfect secrece (Katz & Lindell, 2014). Specifically, OTP can encrypt information using either addition or multiplication. Participants can mask their updates to preserve privacy in FL. A formal OTP scheme usually contains the following two algorithms.

1. $\mathsf{Masking}(x, k) \to \hat{x}$. The masking algorithm takes as input a secret message $x$ and a private key $k$, and it outputs the encryption result $\hat{x}$.
2. $\mathsf{UnMasking}(\hat{x}, k) \to x$. The unmasking algorithm takes as input a encrypted message $\hat{x}$ and a private key $k$, and it outputs the plain message $x$.

Users can apply masking to updates via OTP before uploading to the central servers for aggregation. SA is designed not only to effectively prevent centralized servers from snooping on individual models, but also to defend against attacks from malicious participants and ensure the robustness of the entire FL system. Researchers have proposed several variants of SA to address different threat models and system requirements. We focus on masking-based aggregation schemes. Specifically, there is a set of users $\mathcal{U}$ where $u_i \in \mathcal{U}$ has a private update $x_i$ in FL. In masking-based SA, each $u_i$ adds a pair-wise additive mask to its private update $x_i$ to get the masked vector $y_i$ as follows:

$$y_i = x_i + \sum_{u_j \in \mathcal{U}: i < j} \mathrm{PRG}(s_{i,j}) - \sum_{u_j \in \mathcal{U}: i > j} \mathrm{PRG}(s_{j,i}), \tag{4}$$

where the pseudorandom generator (PRG) can randomly generate a sequence numbers based on the random seed $s_{i,j}$. Note that the masks will be removed when all masked input updates $y_i$ are summed, resulting in

$$\sum_{u_i \in \mathcal{U}} y_i = \sum_{u_i \in \mathcal{U}} \left( x_i + \sum_{i < j} \mathrm{PRG}(s_{i,j}) - \sum_{i > j} \mathrm{PRG}(s_{j,i}) \right) = \sum_{u_i \in \mathcal{U}} x_i. \tag{5}$$

In addition, in order to deal with dropped users during protocol execution, the Shamir secret sharing scheme (Shamir, 1979) is used to share seeds among users. The Diffie-Hellman (DH) key exchange protocol (Diffie & Hellman, 1976) is used to negotiate the seeds $s_{i,j}$ for each pair of users $(u_i, u_j) \in \mathcal{U}$. Note that for large-scale FL applications, the above scheme is not cost-effective. For a $n$-user FL system, it takes $\mathcal{O}(n^2)$ communication rounds to run the pairwise DH key exchange protocol.

## 2.3 MODEL INCONSISTENCY ATTACKS

A malicious server $\mathcal{AS}$ intends to obtain private information about the model update of target user $U_{tar}$. It can distribute elaborately constructed parameters $\theta_{i,t}$ to the non-target users $\{\mathcal{U} \setminus U_{tar}\}$ and then send normal parameters $\theta_{tar,t}$ to the target user, where $\mathcal{U}$ denotes the set of all users. This can trigger the *dying-ReLU* (Lu et al., 2019), where the dead layer cannot generate any gradient. Therefore, the non-target user ends up generating tampered model updates $\Delta_{D_{i,t}}^{\theta_{i,t}}$, where the $D_{i,t}$ is the local date of $U_i$. While the parameters of $U_{tar}$ are real thus generating a right update $\Delta_{D_{tar,t}}^{\theta_{tar,t}}$ on its local data $\mathcal{D}_{tar,t}$ in round $t$. These tampered model updates can enable $\mathcal{AS}$ to obtain the model

updates $\Delta_{D_{tar,t}}^{\theta_{tar,t}}$ of $\hat{U}_{tar}$ in plaintext. Specifically, the final result of secure aggregation is as follows,

$$\mathcal{AS}^{SA}(\Delta_{D_{1,t}}^{\theta_{1,t}}, ..., \Delta_{D_{i-1,t}}^{\theta_{i-1,t}}, \Delta_{D_{tar,t}}^{\theta_{tar,t}}, \Delta_{D_{i+1,t}}^{\theta_{i+1,t}}, ..., \Delta_{D_{n,t}}^{\theta_{n,t}}) \tag{6}$$
$$= \mathcal{AS}^{SA}(0, ..., 0, \Delta_{D_{tar,t}}^{\theta_{tar,t}}, 0, ..., 0) = \Delta_{D_{tar,t}}^{\theta_{tar,t}}.$$

Once $\mathcal{AS}$ gets the update $\Delta_{D_{tar,t}}^{\theta_{tar,t}}$, it can get sensitive information about $\mathcal{D}_{tar,t}$ by executing any gradient inversion attack or inference attacks.

## 3 PROPOSED METHODS

In this section, we design the Janus, a generic privacy-enhanced multi-round SA scheme via a dual-server architecture, where SHC is the core cryptography for verifiability. To facilitate understanding, we first present the new primitive SHC, followed by elaborating on the construction of Janus. Let $\bigodot$ denote the consecutive operation of $\odot$. Specifically, $\bigodot_{i=1}^{n} x_i = x_1 \odot x_2... \odot x_n$, where the $\odot$ indicates addition or multiplication depending on the specific scheme. $T$ is the total number of rounds required for the model to converge and $t$ denotes current round. Let $n$ users participate in FL training, where users are denoted by $\mathcal{U}_t = \{U_i, i \in [1, n]\}$. All users negotiate a model architecture and train the model locally on their private data sets $\mathcal{D}_i$. There are three types of entities in our system which are aggregation server $S_0$, assistant server $S_1$, and users. We assume that each user $U_i \in \mathcal{U}_t$ holds a private update $x_i$ of dimension $m$. For simplicity, we assume that the elements of $x_i$ and $\sum_{U_i \in \mathcal{U}} x_i$ are in $\mathbb{Z}_R$ for $R$.

### 3.1 SEPARABLE HOMOMORPHIC COMMITMENT

**Definition 1** *(Separable Homomorphic Commitment). A secure separable homomorphic commitment scheme is a cryptographic protocol that enables secure and flexible commitments. It is comprised of a set of algorithms denoted by the tuple* (Setup, Commit, Se, PCommit, Reveal). *The formal syntax of each algorithm is described as follows:*

- $pp \leftarrow \mathsf{Setup}(1^\lambda)$. A $\mathcal{PPT}$ initialization algorithm takes as input a security parameter $\lambda$, and it outputs a public parameters $pp$.

- $c \leftarrow \mathsf{Commit}(pp, m, r)$. A $\mathcal{PPT}$ commitment algorithm takes as input a public parameter $pp$, a message $m$ and a random number $r$, and it outputs a complete commitment $c$, where $c = (c_m, c_r)$ and $c_m$ is the part associated with the message $m$ and $c_r$ is related to the random number (blinder) $r$.

- $c_m \leftarrow \mathsf{Se}(pp, c, c_r)$. A Decisional Polynomial Time ($\mathcal{DPT}$) separation algorithm takes as input a public parameter $pp$, a complete commitment $c$ and a blinder-related part $c_r$, and it outputs the message-related commitment $c_m$.

- $c_m \leftarrow \mathsf{PCommit}(pp, m)$. A $\mathcal{DPT}$ commitment algorithm takes as input a public parameter $pp$, a message $m$, and it outputs the message-related commitment $c_m$.

- $1/0 \leftarrow \mathsf{Reveal}(pp, c, m, r)$. A $\mathcal{DPT}$ revealing commitment algorithm takes as input the public parameter $pp$, the complete commitment $c$, the message $m$ and the random blinder $r$, this algorithm outputs 1 if the $m$ is the valid committed message of $c$ and 0 otherwise.

In addition to the completeness, binding and hiding properties possessed by traditional commitment schemes described in Section 2, the SHC also possess the following two unique properties. The two servers independently aggregate the different values in the commitments. SHC is able to separate part of the message and compare it with the aggregated results, thereby ensuring the correctness of the aggregation.

- **Separability.** The complete commitment $c$ generated by $\mathsf{Commit}(m, r)$ can be divided into two parts $c = (c_m, c_r)$, where $c_m$ is the part associated with the commitment message $m$ and $c_r$ is related to the random blinder $r$. It can use $c_r$ to extract from the complete commitment $c$ only the parts that are relevant to $m$. Taking the classic Pdeersen commitment (Pedersen, 1991) as an example, the complete commitment is $c = h^r g^m$. Given

| Users | $S_0$ | $S_1$ |
|---|---|---|

$\hat{x}_{i,t} \leftarrow \text{Masking}(x_{i,t}, sk_{i,t})$
$CT_{i,t} \leftarrow \text{Enc}(pk_s, sk_{i,t})$
$(c_{i,t}, c_{i,r}) \leftarrow \text{Commit}(x_{i,t}, r_{i,t})$

$(c_{i,r}, \hat{x}_{i,t})$ $\qquad\qquad\qquad$ $(c_{i,t}, CT_{i,t})$

$\hat{X}_t = \odot_{i=1}^n \hat{x}_{i,t}$
$C_r = \odot_{i=1}^n c_{i,r}$

$(\hat{X}_t, C_r)$

$sk_{i,t} \leftarrow \text{Dec}(sk_s, CT_{i,t})$
$SK_t = \odot_{i=1}^n sk_{i,t}$
$C_t = \odot_{i=1}^n c_{i,t}$

$(SK_t, C_t)$

$X_t \leftarrow \text{UnMasking}(\hat{X}_t, SK_t)$
$C_m \leftarrow \text{Se}(C_t, C_r)$
$C_m^* \leftarrow \text{PCommit}(X_t, pp_c)$
$C_m \overset{?}{=} C_m^*$

Figure 1: The Workflow of Janus.

$c_r = h^r, c_m = g^m$, we can get the $c_m$ from $c$ and $c_r$ via $c/c_r$. Furthermore, the $c_m$ can be calculated from $\text{PCommit}(m, pp)$.

- **Homomorphism.** Homomorphism facilitates to accomplish secure aggregation. Define the space of message, blinder and commitment as $\mathcal{M}_c, \mathcal{R}_c, \mathcal{C}_c$ respectively.

$$\forall (m_0, r_0), (m_1, r_1) \in \mathcal{M}_c \times \mathcal{R}_c :$$
$$\text{Commit}(m_0 + m_1; r_0 + r_1) = \text{Commit}(m_0; r_0) \cdot \text{Commit}(m_1; r_1). \tag{7}$$

### 3.2 THE PROPOSED JANUS

Janus tackles the challenges of dynamic user participation, verifiability, and resistance to model inconsistency attacks that are not addressed in the state-of-the-art Flamingo (S&P'23). Specifiaclly, it has following three key high-level technical ideas:

(1) Dual-server architecture and dynamic user participation. Specifically, the Janus involves two servers, $S_0$ and $S_1$. $S_0$ is responsible for aggregating the masked updates and $S_1$ is responsible for aggregating the values associated with the commitments. The dual-server architecture prevents the servers from accessing the final aggregation results, thus effectively avoids attacks such as model reversal and model inconsistency, which are serious privacy leakage in traditional single-server. Furthermore, there is no need to re-establish complex communication diagrams when users join or leave. New users can participate in the new training process by simply generating their own public/private keys and obtaining the servers' public keys.

(2) Lightweight components and efficient aggregation. Instead of requiring the client to secretly share the mask with all its neighbours as Flamingo and BBSA, Janus does not even require neighbours and avoids the time-consuming process of negotiating keys with each other. It only applies OTP to mask the secret updates and subsequently encrypts the masks via a secure public key encryption. The different messages are then sent to $S_0$ and $S_1$. Thus, no matter how the number of users in the system increases, the operations required by Janus are fixed to the desired constant level.

(3) Verifiability and privacy enhancement. The separability of SHC allows the user to validate the aggregated values locally, thus enabling verifiability. In addition, the binding feature of SHC prevents the client from denying previously sent malicious messages when subsequent misbehavior is detected. This is a feature not available in other advanced schemes. Given the hiding of the SHC and the confidentiality of public key encryption, neither $S_0$ nor $S_1$ can access the received secret information. Combined with our dual-server architecture, higher security can be achieved.

Figure 1 shows the workflow of Janus. Subsequently, we provide a detailed description of our Janus, noting that it is a generic construction. Thus, we assume the underlying public key encryption scheme is $\Pi_E = (\text{Setup}, \text{KeyGen}, \text{Enc}, \text{Dec})$, the OTP scheme is $\Pi_O = (\text{Masking}, \text{unMasking})$, and the SHC scheme is $\Pi_S = (\text{Setup}, \text{Commit}, \text{Se}, \text{PCommit}, \text{Reveal})$, in which the setup parts of these schemes are completed in the *Setup* phase of Janus by default. Furthermore, Appendix B

gives the tasks of the different entities in each phase for conciseness and an effective instantiation to demonstrate the practicality. Specifically, Janus consists of the following four phases:

**Setup.** The objective of this phase is to determine the public parameters $pp$ and specific cryptographic schemes, which ensures that subsequent schemes work properly. In round $t$, all parties are given the security parameter $\lambda$. All public parameters $pp$ of the system are then generated based on $\lambda$, e.g., the setup phase and public parameters generation in $\Pi_E, \Pi_O, \Pi_S$. Each user will generate their private key $sk_{i,t}$ for the OTP. The $S_1$ will generate its public/private key $(pk_s, sk_s)$ and publish its public key to all participants. Subsequent communications between the users and the servers are encrypted with their respective public keys by default.

**Masking and Report.** The $U_i$ masks its input updates $x_{i,t}$ via $\mathsf{Masking}(x_{i,t}, sk_{i,t})$ to get the masked updates $\hat{x}_{i,t}$. Subsequently, $U_i$ encrypts the $sk_{i,t}$ using the public key of $S_1$ via $\mathsf{Enc}(pk_s, sk_{i,t})$ to get the ciphertext $CT_{i,t}$ of $sk_{i,t}$. To achieve subsequent verifiability, $U_i$ makes separable commitment for the input updates $x_{i,t}$ via $\mathsf{Commit}(x_{i,t}, r_{i,t})$ to get the full commitment $c_{i,t}$, where the $r_{i,t}$ is the blinder, the $c_{i,t}$ can be divided into $(c_{i,r}, c_{i,m})$, $c_{i,r}$ is the commitment of blinder and $c_{i,m}$ is the commitment of updates. Then it sends $(\hat{x}_{i,t}, c_{i,r})$ to the aggregation server $S_0$ and $(c_{i,t}, CT_{i,t})$ to the assistant server $S_1$.

**Collection and Aggregation.** In this phase, the servers will complete the computation secure aggregation and verification for users updates. Specifically, $S_0$ will aggregate the masked input updates from all users via $X_t = \bigodot_{i=1}^n \hat{x}_{i,t} = \hat{x}_{1,t} \odot \hat{x}_{2,t} \odot ... \odot \hat{x}_{n,t}$. Then $S_0$ computes $C_r = \bigodot_{i=1}^n c_{i,r} = c_{1,r} \odot c_{2,r} \odot ... \odot c_{n,r}$. $S_0$ sends $(\hat{X}_t, C_r)$ to all users. In fact, $\hat{X}_t$ contains the updated aggregated values for round $t$ and $C_r$ can assist in the validation of aggregated result. For the $S_1$, it first decrypts the ciphertext to get the $sk_{i,t}$ via $\mathsf{Dec}(sk_s, CT_{i,t})$. Then it can aggregate the $\bigodot_{i=1}^n sk_{i,t} = sk_{1,t} \odot sk_{2,t} \odot ... \odot sk_{n,t} = SK_t$. Furthermore, it calculates the aggregation result of the full commitment value for subsequent users to verify the aggregation result completed by $S_0$ via $\bigodot_{i=1}^n c_{i,t} = c_{1,t} \odot c_{2,t} \odot ... \odot c_{n,t} = C_t$. Finally, $S_1$ sends $(SK_t, C_t)$ to all users.

**UnMasking and Verification.** The users compute the final update results based on the values returned by the two servers and validate the aggregated result. Specifically, $U_i$ gets the final aggregation result via $X_t = \mathsf{UnMasking}(\hat{X}_t, SK_t)$, where the $X_t$ is the updates aggregation result of the round $t$. To verify the correctness of the aggregation result, $U_i$ extracts the commitment value related to the updates via $C_m = \mathsf{Se}(C_t, C_r)$. The user then calculates the commitment value which is only related to the updates via $C_m^* = \mathsf{PCommit}(X_t, pp_c)$, where the $pp_c$ is the public parameter of the underlying SHC. Finally, $U_i$ compares whether $C_m^*$ and $C_m$ are equal. If they are equal, then the aggregated result is correct; otherwise, it is invalid, and $U_i$ will terminate the subsequent training.

# 4 EVALUATION

## 4.1 THEORETICAL ANALYSIS

Janus offers enhanced security compared to state-of-the-art schemes. We give a formal security analysis in appendix C, where Janus can resist MIA and achieve multi-round security. Furthermore, a key advantage of Janus over Flamingo and BBSA is its ability to complete each round with fewer interactions. The two advanced schemes necessitate communication with neighboring nodes to complete the elimination of the mask or decryption process. Assuming that the underlying operations, such as commitments and encryptions, have a complexity of $\mathcal{O}(1)$, Janus demonstrates superior efficiency in terms of interaction count. The remarkable property of Janus is that the system overhead do not grow with the number of users as in previous schemes. The system is designed to be client-friendly, minimizing computational overhead. Clients need only two interactions with the servers to go offline, ensuring there are no issues with aggregation failures or inaccurate results due to user disconnection. We focus on a round of aggregation, with Table 2 presenting the results in comparison to relevant advanced schemes.

*Computation Cost.* The computation cost of each client consists of: 1) masking the local update by using one-time pad; 2) encrypting the key of one-time pad by public key encryption; 3) committing the local update by using the SHC; 4) unmasking the global aggregation result; 5) separating message-only commitments from the full commitment; 6) calculating the commitment value based on the unmasking result and compare whether it is equal to the separated commitment value to com-

Table 2: Comparison of Performance Analysis

| Scheme | Computation | | Communication | |
|---|---|---|---|---|
| | Client | Server | Client | Server |
| SecAgg | $\mathcal{O}(n^2 + md)$ | $\mathcal{O}(dn^2))$ | $\mathcal{O}(n + m)$ | $\mathcal{O}(n^2 + mn)$ |
| BBSA | $\mathcal{O}(A^2 + lA)$ | $\mathcal{O}(n(A^2 + lA))$ | $\mathcal{O}(A^2 + l)$ | $\mathcal{O}(n(A^2 + l))$ |
| VeriFL | $\mathcal{O}(n)$ | $\mathcal{O}(n + l)$ | $\mathcal{O}(n)$ | $\mathcal{O}(1) + \mathcal{O}(n)$ |
| ELSA | $\mathcal{O}(1 + l)$ | $\mathcal{O}(n + nl)$ | $\mathcal{O}(1)$ | $\mathcal{O}(n)$ |
| Flamingo | Regular Client: $O(L^2)$
Decryptor: $\mathcal{O}(L^2 + \delta An + (1 - \delta)n + \epsilon n^2)$ | $\mathcal{O}(n + L^2)$ | Regular Client: $\mathcal{O}(l + A + L^2)$
Decryptors: $\mathcal{O}(L^2 + L + \delta An + (1 - \delta)n)$ | $\mathcal{O}(L^3 + n(l + L + A))$ |
| Janus | $\mathcal{O}(1 + l)$ | $\mathcal{O}(n + nl)$ | $\mathcal{O}(1)$ | $\mathcal{O}(n)$ |

$^*$ Let $n, L, A$ denote the total number of clients, the number of decryptors and the upper bound number of neighbors of a client respectively, where $A = \log n$ in BBSA. $l$ denotes the dimension of the update. $\delta$ denotes the dropout rate respectively. $\epsilon$ is the parameter of graph generation.

plete the verification. All the above operations take only $O(1)$ time each. Overall, the computational overhead of each client is constant. The computation cost of $S_0$ mainly consists of aggregating the masking updates from clients and the commitment of random numbers, which both take $O(n)$. Thus the total computational overhead grows linearly with the number of clients. For $S_1$, the computation cost consists of: 1) decrypting the ciphertext of the private key of one-time pad; 2) aggregating the private keys for masking; 3) aggregating the complete commitments for subsequent verification of the aggregation result of $S_0$. All these operations mentioned above take $O(n)$. Overall, the communication overhead of servers grows linearly with the number of clients which takes $O(n)$.

*Communication Cost.* Each client needs to send one masked message to $S_0$, one encrypted and committed message to $S_1$. Overall, the computational overhead of each client is constant. For the servers, $S_0$ will send the aggregation result of the masking updates to all clients, which takes $\mathcal{O}(n)$. $S_1$ sends the aggregation result of the key used for one-time pad and the full commitment to all clients, which also takes $O(n)$. Overall, for servers, their communication overhead grows linearly with the number of clients which takes $\mathcal{O}(n)$.

### 4.2 Model Performance

In this section, we carried out various experiments to verify the effectiveness and efficiency of our scheme and to compare it with similar advanced schemes. Our experimental setup includes a 13th Gen Intel(R) Core(TM) i7-13700KF 3.40 GHz processor with 32.0 GB of RAM, a 64-bit Windows 11 operating system, and an RTX 4070Ti GPU display adapter.

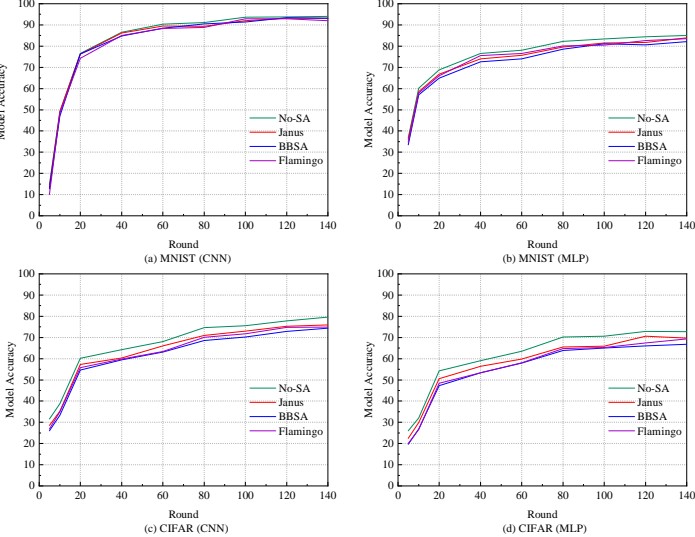

Figure 2: Test accuracy across different datasets and models.

**Baselines.** To evaluate the impact of SA on federated learning (e.g., training effectiveness, communication time), we implemented the original FL framework (No-SA), where the server aggregates clear updates from users in each training round (McMahan et al., 2017). Bell et al. (2020) optimized the communication graph of the first mask-based SA scheme (Bonawitz et al., 2017) and proposed an advanced scheme (BBSA), which we implemented for comparison. For client dropout, we construct the graph with responsive clients, yielding better results than the original. Flamingo (Ma et al., 2023) introduced multi-round aggregation, and both Flamingo and BBSA involve waiting for messages from at least $t$ out of $n$ clients.

**Datasets and Models.** MNIST consists of 70,000 grayscale handwritten digit images (60,000 for training, 10,000 for testing), each 28x28 pixels. We use 100 clients, each with 600 training samples. The global model for MNIST is a fully connected network with layers of size (784, 256, 10). CIFAR-10 includes 60,000 color images across 10 classes (50,000 training, 10,000 testing), using a CNN architecture with a batch size of 10, learning rate of 0.001, and 100 training epochs. We employed SGD as the optimizer, with each client applying SGD once per global epoch (local epoch = 1).

To comprehensively evaluate the impact of the security SA in this paper on the model training effectiveness, our experiments are carried out on different datasets and models. We conducted the training with 100 clients and compare the test accuracy of our Janus with related schemes. Figure 2 shows the comparison results. The following conclusions can be drawn from the experimental results. Firstly, the final test accuracy at model convergence is not much different between our scheme and the compared schemes, in which No-SA has the highest accuracy, and our scheme follows closely.

Specifically, for MNIST, the test accuracy of No-SA can reach to 94.1% under the CNN, while the Janus can also reached about 93.18%. Additionally, the test accuracy of No-SA can reach to 85.04% under the MLP, while the Janus can also reached about 83.95%. Compared to other schemes, Janus has considerable accuracy. As for the CIFAR, the test accuracy of No-SA can reach to 77.8% under the CNN, while the Janus can also reached about 75.94%. Additionally, the test accuracy of No-SA can reach to 72.8% under the MLP, while the Janus can also reached about 71.6%.

Figure 3 shows the loss of related schemes during the training process with different datasets and models. It can be concluded that as the number of training rounds increases, the loss values for the same dataset with different secure aggregation schemes applied are smoother and eventually all converge to be almost equal. This shows that our Janus, like advanced schemes, does not result in a loss of model performance due to the use of secure aggregation. The impact on the model is similar to that of existing advanced schemes, while protecting users privacy and providing better efficiency.

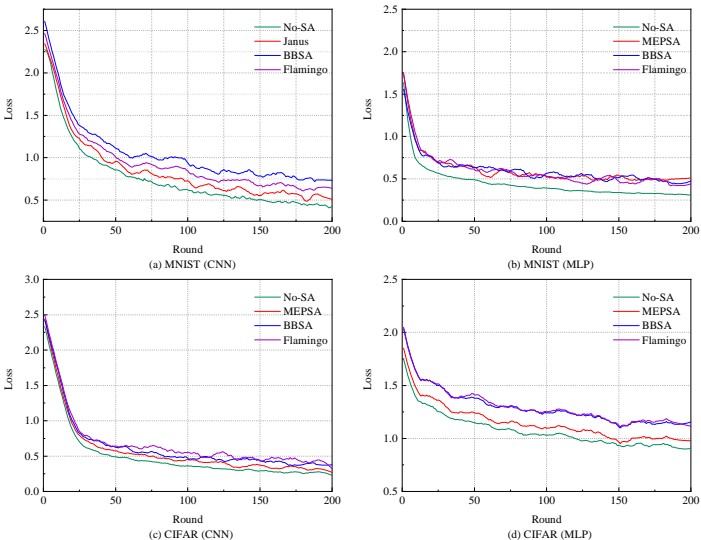

Figure 3: Training loss across different datasets and models.

## 4.3 COMPUTATION OVERHEAD

Since masking-based schemes are not resistant to user dropouts, we consider this case when implementing BBSA and Flamingo. Specifically, we only consider the case where 10% of users drop out, but it should be noted that in practice the waiting time required to solve the user dropout problem is much longer than that considered in our experiments, due to the complexity and diversity of the real scenarios. Moreover, it is important to note that some of the evaluated schemes inherently support multi-round aggregation, while others do not. We adapted the schemes that lack built-in multi-round aggregation capabilities by running them multiple times to simulate the effect of multi-round aggregation. Although this approach is feasible, it introduces a considerable amount of additional and unnecessary computationa overhead. This further highlights the advantages of our proposed scheme, Janus, which is natively designed to support multi-round aggregation without incurring such overhead, thus demonstrating superior efficiency and scalability in practice.

As shown in Figure 4, we present a comparative analysis of the time overhead of various schemes, focusing on the completion time required for a single aggregation. It should be noted that, due to differences in the stages involved across these schemes, only the relevant time-consuming stages were considered for each. From the results, several conclusions can be drawn. First, the computational overhead introduced by SA is within an acceptable range, demonstrating its practicality in real-world applications. More importantly, our proposed scheme exhibits significantly lower overhead, particularly on the client side, which substantially enhances overall efficiency. This improvement can be attributed to the adoption of lightweight cryptographic components, which circumvent time-intensive operations such as secret sharing and DH key negotiation. The absence of these complex operations reduces the computational burden on clients, thereby contributing to the superior performance of our scheme.

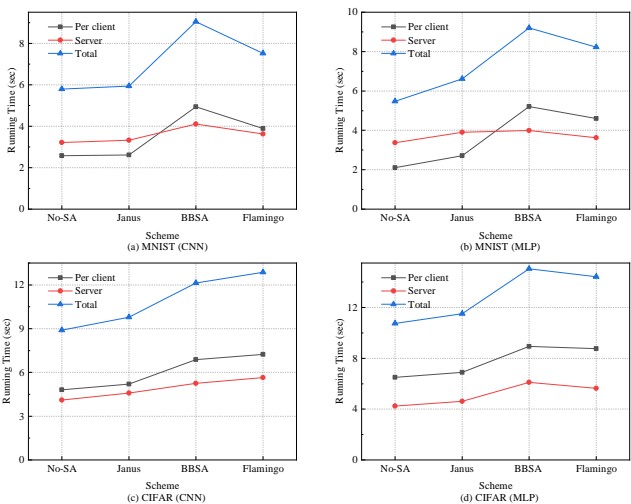

Figure 4: Computation overhead across different datasets and models.

## 5 CONCLUSION

In this paper, we propose a new cryptographic primitive, i.e., separable homomorphic commitment and design a generic dual-server multi-round SA scheme called Janus for federated learning. Janus addresses the issues of dynamic user participation, verifiability, and resistance to model inconsistency attacks that are not considered in advanced Flamingo (S&P'23). It not only significantly enhances security but also improves system efficiency, which reduces per-client communication and computation overhead from a logarithmic to a constant scale compared to current state-of-the-art methods, with almost no compromise in model accuracy. Finally, we evaluate Janus from both theoretical and experimental perspectives, demonstrating its superior security and performance. Future researches on integrating Janus with various advanced privacy-preserving techniques could further enhance its security. Additionally, secure and effective identification of data poisoning attacks from the users is another worthwhile research direction.

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

## A  RELATED WORKS

The main goal of FL is to protect the privacy of local data while still allowing it to be used to train the public models. A significant amount of research has been conducted around SA. This section reviews the work related to our scheme. Refer to the reference (Qi et al., 2024; Liu et al., 2022) for more extensive survey in this field.

**Masking-based SA.** Masking is a classic encryption technique based on one-time pad (Katz & Lindell, 2014). Bonawitz et al. (2017) designed the first SA scheme (SecAgg) which using pairwise masks to hide individual inputs for FL. However, their scheme involves a complete communication graph, which incurs heavy computation and communication for each client linear in the number of participants. Subsequently, Bell et al. (2020) replaced that with a $k$-regular graph of logarithmic degree, which greatly improved the efficiency while maintaining the security. Stevens et al. (2022) replaced the standard mask with learning with errors mask and used verifiable secret sharing to prevent malicious users from distributing incorrect shares. Sandholm et al. (2021) arranged the users in the system in the form of a ring chain, the efficiency of the scheme has been significantly improved, and the user drop problem can be effectively solved. Most masking-based schemes require double masking in order to solve the problem of dropped users. Bonawitz et al. (2019) combined the random rotation technique to actively adjust the quantisation range of the model in order to reduce the model volume. To reduce the communication overhead, TurboAgg (So et al., 2021) divides $n$ users into $n/\log n$ groups and then uses a multi-group loop structure for subsequent aggregation.

**Attacks that Bypass SA.** The aggregation results of most existing schemes are visible to both the clients and the aggregation server. However, this can lead to attacks where malicious servers bypass the SA. Pasquini et al. (2022) proposed model inconsistency attacks, where a malicious server can distribute different parameters to targeted and non-targeted users. This can trigger *dying-ReLU* and make the input of non-target users be zero. So et al. (2023) noticed that when the trained model begins to converge, the client model changes little between one training step and the next. A malicious server can infer the updates of a client that participated in the previous round but did not participate in the subsequent round from the aggregation results. Gao et al. (2023) proposed a scheme which can launch a category inference attack even in presence of SA. To avoid this type of attack, when the clients receive the model parameters, they need to verify whether the received parameters are consistent or not, and terminate the training if they are not. But this will increase the system overhead. Fernández et al. (2021) applied differential privacy on the aggregated model to hide the aggregation results.

**Server-side Attacks and Defenses.** Membership inference attacks pose a potential threat from the server side in FL. Specifically, an adversary can determine whether some specific data records are part of the local training dataset of a target user only by accessing the model updates, either through a black-box or white-box approach. Yeom et al. (2018) proposed the first label-based attack, which aims at predicting whether an instance is in the local data of the target user. The attacker exploits the performance disadvantage of the target model in the test dataset to complete this attack. Chen & Vikalo (2024) proposed a general analytical method that allows the FL server to recover client training labels, applicable to various FL algorithms without assumptions on activation functions or batch label composition. Shokri et al. (2017) designed an attack with partial output knowledge in a black box-scenario. Furthermore, Salem et al. (2019) improved a new attack by using the maximum value of the model output confidence. Zhuang et al. (2024) introduced the layer substitution analysis, a new technique that identifies layers critical for backdoor injection, making it well-suited for FL attacks. Leveraging this technique, they developed two layer-wise backdoor attack strategies that successfully implant backdoors into these key layers and evade state-of-the-art defenses without compromising the primary task accuracy.

Meanwhile, Bonawitz et al. (2017) proposed the first SA scheme to compute the sum of model updates hiding personal information. Subsequently, a great deal of research has centred around SA. Techniques such as homomorphic encryption (Zhang et al., 2020a), differential privacy (Stevens et al., 2022), and multi-party computation (Bell et al., 2020) are used to construct SA schemes to protect user privacy from attack by malicious servers. SA based on cryptography aims to prevent attacks by concealing model updates from any potential adversaries. This approach ensures that individual contributions remain private, making it difficult for malicious entities to infer sensitive information from the data.

Recently, Xie et al. (2024) identify a limitation in existing model poisoning attacks defenses: reliance on cross-client or global information, which leads to performance degradation under non-IID data distributions or when there is a large number of malicious clients. Then they establish a crucial distinction between model poisoning attacks and benign model updates by determining whether the update can be approximately reconstructed using distilled local knowledge. Wu et al. (2024) proposed FedInverse, a framework designed to evaluate whether FL models are susceptible to model inversion attacks and quantify the associated data-leakage risks. Garov et al. (2024) showed that all existing malicious server attacks can be identified through systematic checks. Furthermore, they established a set of essential requirements that any practical malicious server attack must meet.

**Verifiability.** In addition, a malicious server might return incorrect aggregation results to gain an unfair advantage or disrupt the system's integrity. Such behavior poses significant security threats, as users or clients relying on these results could be misled or manipulated. Therefore verifiable SA is necessary to ensure correct aggregation. Zhang et al. (2020b) verified the aggregation result via homomorphic encryption SA using homomorphic hash function. Additionally, Xu et al. (2020) verified masking-based SA using the same technique. Guo et al. (2021) proposed a verification scheme which focuses on the high dimension inputs. Brunetta et al. (2021) proposed a non-interactive verifiable SA protocol from NIVA, which requiers users create a tag for each input shares. In contrast, Tsaloli et al. (2021) proposed a scheme requires only a single tag for each user.

**Multi-round Setting and Dynamic Joining.** Model convergence in Federated Learning (FL) typically requires multiple rounds of training, with each round contributing incrementally to the overall performance of the global model. However, most existing state-of-the-art SA schemes are designed to support only a single round of aggregation. In addition to protecting user privacy in single rounds of FL training, some studies have looked at privacy issues arising from multiple rounds of FL training. Nguyen et al. (2022) and So et al. (2022) proposed two new schemes support asynchronous aggregation. Guo et al. (2022) designed a multi-round SA protocol for reusable secrets, and their scheme is mainly oriented towards scenarios with small inputs (the input vector with small values).

Recently, Ma et al. (2023) proposed Flamingo, which has no restrictions on input value. So et al. (2023) mitigated the privacy leakage involved in multi-round aggregation through client selection. Furthermore, the existing schemes do not support dynamic joining. Flamingo assumes that the set of all clients $(n)$ participating in the training is fixed before the training starts and some subset is selected from $n$ in each round $t$. Therefore, Flamingo does not support the user to dynamically add in the training process. Most current schemes require reconstruction of the communication graph when new users join and require key negotiation with each other user, which imposes huge

1. **Setup.**

   – All parties get the security parameter $\lambda$. – This phase generates the public parameter $pp$ of the system, which contains the specific commitment, one-time pad and public key encryption.

   – The assitant server $S_1$ generates its public/private key $(pk_s, sk_s)$ and publishes its public key to all users.

   – Each user generates its public/private key $(pk_i, sk_i)$ and publish its public key to servers $S_0$ and $S_1$. (Subsequent user-server interactions via public key encryption by default.)

2. **Masking and Report.**

   – Each user computes $\hat{x}_{i,t} \leftarrow \mathsf{Masking}(sk_{i,t}, x_{i,t})$, where the $sk_{i,t}$ is the private key generated by user $U_i$ during the round $t$.

   – Each user encrypts the private key in OTP $CT_{i,t} \leftarrow \mathsf{Enc}(pk_s, sk_{i,t})$, where the $pk_s$ is the public key of the assistant server $S_1$.

   – Each user generates the commitment $c_{i,t} = (c_{i,r}, c_{i,m}) \leftarrow \mathsf{Commit}(x_{i,t}, r_{i,t})$, where the $r_{i,t}$ is the blinder and $c_{i,r}, c_{i,m}$ is the the commitment parts of blinder and message respectively.

   – Each user sends $(\hat{x}_{i,t} \| c_{i,r})$ to $S_0$ and $(CT_{i,t} \| c_{i,t})$ to the $S_1$.

3. **Collection and Aggregation.**

   – $S_0$ collects the messages $(\hat{x}_{i,t} \| c_{i,r})$ from users and parses as $x_{i,t}$ and $c_{i,r}$.

   – Then $S_0$ computes the $\bigodot_{i=1}^{n} \hat{x_{i,t}} = \hat{X}_t$ and $\bigodot_{i=1}^{n} c_{i,r} = C_r$.

   – $S_0$ sends the $\hat{X}_t$ and $C_r$ to all users.

   – $S_1$ collects the messages $(CT_{i,t} \| c_{i,t})$ from users and parses as $CT_{i,t}$ and $c_{i,t}$.

   – $S_1$ decrypts the $sk_{i,t} \leftarrow \mathsf{Dec}(CT_{i,t}, sk_s)$ and it computes $\bigodot_{i=1}^{n} sk_{i,t} = SK_t$.

   – $S_1$ computes the $\bigodot_{i=1}^{n} c_{i,t} = C_t$.

   – $S_1$ sends the $SK_t$ and $C_t$ to all users.

4. **UnMasking and Verification.**

   – Each user receives the message from $S_0$ and $S_1$, then it decrypts the ciphertext as $C_r$ and $\hat{X}_t$ using its private key $sk_i$.

   – Each user unmasks the aggregation $X_t \leftarrow \mathsf{UnMasking}(SK_t, \hat{X}_t)$.

   – Each user computes the commitment about the input updates $C_m \leftarrow \mathsf{Se}(C_t, C_r)$.

   – Each user generates the commitment of $C_m^* \leftarrow \mathsf{PCommit}(X_t, PP_c)$, which is related to the updates. $PP_c$ is the public parameter of commitment scheme. Then $U_i$ compares $C_m^* \stackrel{?}{=} C_m$. If it is equal, then the aggregation result completed by $S_0$ is correct, otherwise it is invalid. Once the aggregation results are found to be incorrect, the user terminates the subsequent training.

Figure 5: Detailed Construction of Janus.

communication and computation overheads. In addition, Wang et al. (2024) focus on the aggregation of cross-round local models. They proposed FedCDA, a novel cross-round aggregation method that constructs the global model by aggregating local models from multiple rounds based on minimum divergence. To enhance efficiency, FedCDA further introduces an approximation strategy to reduce selection overhead.

## B  DETAILED JANUS AND ITS INSTANTIATION

In this section, Figure 5 gives the full generic construction of Janus. Furthermore, we give an effective instantiation of our generic construction, where the underlying SHC is Pedersen commitment,

public key encryption is ElGamal, one-time pad is based on normal addition encryption. Specifically, our scheme consists of the following five phases: setup, masking and report, collection and aggregation, collection and aggregation, unmasking and verfication.

*Setup.* This phase determines the public parameters of the system. Firstly, all participants agree on the security parameter $\lambda$. The public parameters of the cryptographic primitives are then generated based on the security parameter. Define a triplet $(p, q, g, h)$, where $p$ is a randomly chosen prime of length $|q| = \lambda + \delta$, the $\delta$ is a specified constant, $q$ is a prime order group of $\mathbb{Z}_p^*$, and $g, h$ are random generators of group of $q$ order, $q = (p - 1)/\gamma$ is prime and the $\gamma$ is a specified small integer. $U_i$ generates public/private key $(sk_i, pk_i) = (sk_i, g^{sk_i} \pmod{p})$, where the $sk_i \in \mathbb{Z}_p^*$. $S_1$ generates public/private key $(sk_s, pk_s) = (sk_s, g^{sk_s} \pmod{p})$ where the $sk_s \in \mathbb{Z}_p^*$. Then $S_1$ and $U_i$ publish their public keys to all entities while store their private keys secretly.

*Masking and Report.* Each user $U_i$ trains local data $\mathcal{D}_i$ to get the updates $x_{i,t}$ for round $t$. $U_i$ masks the vector by $\hat{x}_{i,t} = \mathsf{Masking}(x_{i,t}, sk_{i,t}) = x_{i,t} + sk_{i,t} \pmod{p}$. Then $U_i$ encrypts the $sk_{i,t}$ by $\mathsf{Enc}(pk_s, sk_{i,t}) = CT_{i,t} = (g^{k_{i,t}} \pmod{p}, sk_{i,t}pk_s^{k_{i,t}} \pmod{p})$. Furthermore, $U_i$ commits the $x_{i,t}$ by $c_{i,t} = \mathsf{Commit}(x_{i,t}, r_{i,t}) = g^{x_{i,t}}h^{r_{i,t}} \pmod{p}$, where the $r_{i,t} \in \mathbb{Z}_p^*$ and $c_{i,t} = (c_{i,r}, c_{i,m}) = (h^{r_{i,t}} \pmod{p}, g^{x_{i,t}} \pmod{p})$. Finally, $U_i$ sends $c_{i,r}$ and $\hat{x}_{i,t}$ to $S_0$, $c_{i,t}$ and $CT_{i,t}$ to $S_1$.

*Collection and Aggregation.* Subsequently, $S_0$ receives the message from $U_i$. Then it computes $\bigodot_{i=1}^n \hat{x}_{i,t} = \hat{x}_{1,t} + \hat{x}_{2,t} + ... + \hat{x}_{n,t} = \hat{X}_t$ and $\bigodot_{i=1}^n c_{i,r} = h^{r_{1,t}}h^{r_{2,t}}...h^{n,t} \pmod{p} = C_r$. Then $S_0$ sends $C_r$ and $\hat{X}_t$ to all users. When the $S_1$ receives the message from $U_i$. It first decrypts $sk_{i,t} = \mathsf{Dec}(sk_s, CT_{i,t}) = sk_{i,t}pk_s^{k_{i,t}}(g^{k_{i,t}^{sk_s}})^{-1} \pmod{p}$. Subsequently, it computes $\bigodot_{i=1}^n c_{i,t} = c_{1,t}c_{2,t}...c_{n,t} \pmod{p} = C_t$. Then it computes $\bigodot_{i=1}^n sk_{i,t} = sk_{1,t} + sk_{2,t} + ... + sk_{n,t} = SK_t$. Finally, $S_1$ sends the $C_t$ and $SK_t$ to all users.

*Unmasking and Verfication.* When $U_i$ receives the message from $S_0$ and $S_1$. Firstly, $U_i$ computes the $X_t = \mathsf{Unmasking}(\hat{X}_t, SK_t) = \hat{X}_t - SK_t$ to get the aggregation result $X_t$. To verify the validity of the aggregation results, $U_i$ separates the parts of the commitments that are only relevant to the input updates by $\mathsf{Se}(C_t, C_r) = C_m$. Then $U_i$ makes a commitment to the aggregation result from $S_0$ through $\mathsf{PCommit}(X_t, pp_c) = g^{\hat{X}_t} \pmod{p} = C_m^*$, where $pp_c$ is the public parameters of the underlying SHC. Eventually $U_i$ compares whether $C_m^* \stackrel{?}{=} C_m$ holds, if it does it indicates that the aggregation result $\hat{X}_t$ from $S_0$ is correct, otherwise the aggregation result is not valid. $U_i$ will refuse to accept the results of the aggregation and aborted the subsequent training.

**Correctness.** The correctness of this instantiation requires each user will obtain the correct aggregation result and the valid verification as long as each entities run the protocol honestly. It is not hard to prove this due to the correctness of the underlying public key encryption, one-time pad and SHC. Specifically, we asume that the aggregation server $S_0$ receives all masked-input and performs Janus correctly, the following condition holds.

$$\bigodot_{i=1}^n \hat{x}_{i,t} = \hat{x}_{1,t} + \hat{x}_{2,t} + ... + \hat{x}_{n,t}$$
$$= x_{1,t} + sk_{i,t} + x_{2,t} + sk_{2,t} + ... + x_{n,t} + sk_{n,t} \qquad (8)$$
$$= \bigodot_{i=1}^n x_{i,t} + \bigodot_{i=1}^n sk_{i,t}$$
$$= X_t + SK_t,$$

where the $\bigodot_{i=1}^n sk_{i,t}$ is computed by $S_1$. The final aggregation result is $\bigodot_{i=1}^n x_{i,t} = \bigodot_{i=1}^n \hat{x}_{i,t} - \bigodot_{i=1}^n sk_{i,t} = X_t$. If the validation passes, the following condition holds.

$$C_t = g^{x_{1,t}}h^{r_{1,t}}g^{x_{2,t}}h^{r_{2,t}}...g^{x_{n,t}}h^{r_{n,t}}$$
$$= g^{x_{1,t}+x_{2,t}+...+x_{n,t}}h^{r_{1,t}+r_{2,t}+...+r_{n,t}},$$
$$C_r = h^{r_{1,t}+r_{2,t}+...+r_{n,t}}, \qquad (9)$$
$$C_m = C_t/C_r = g^{x_{1,t}+x_{2,t}+...+x_{n,t}},$$
$$C_m^* = g^{X_t}.$$

If the aggregation result $X_t$ from $S_0$ is correct, then the $C_m^* = C_m$ will always hold.

## C  SECURITY ANALYSIS

In this section, we intend to demonstrate the security of our generic construction. We first give the threat model and prove the Janus can protect the privacy of users' local updates and the aggretated upates. Finally, we give the security proof of single round and multi-round.

### C.1  THREAT MODEL

All users agree to publish the final results of model aggregation only to each user, but not to the servers to resist MIA. These users have a common interest in soundness (i.e., getting the correct global model aggregation updates from untrusted servers) and privacy (i.e., hiding local model updates from each other and the server). The specific assumptions in our paper are as follows: The two servers will not collude but may perform incorrect aggregation. The scheme also allows for up to $n - 2$ clients to collude. Specifically, even if the server aggregates incorrect results, our scheme provides verifiability, which enables us to detect such behavior and mitigate the associated risks. If the server colludes with up to $n - 2$ clients, it can only obtain the additive result of the remaining two uncolluding clients. This result is an aggregation of two encrypted or obfuscated values, making it impossible to recover each uncolluding user's specific gradient information. This ensures that the colluding entities cannot initiate a MIA or access the private information of the remaining two non-colluding clients. When $n - 2$ clients collude, this assumption is even weaker, as the absence of server involvement further limits the accessible information, making it even harder to extract useful data. If only a single server is corrupted, this does not compromise individual user privacy. For instance, with server $S_0$, as long as the underlying encryption algorithm is secure, the server cannot access the user-submitted private data without the user's private key. Similarly, for server $S_1$, the security of the underlying SHC ensures that its hiding properties prevent $S_1$ from obtaining any private information. In conclusion, the assumptions of our scheme are reasonable and well-supported. We will incorporate these clarifications in the revised version to better highlight the theoretical advantages of our approach. In addition, we assume the channel between each user and servers are secure, which allows each entities to authenticate the incoming messages and prevent outsiders from injectiong their responses. Furthermore, we assume that there is no collusion between all entities in the system. Our security proofs are based on this threat model.

### C.2  PRIVACY FROM USERS

In the "honest but curious" setting, each client will honestly adhere to the protocol but attempt to infer the local gradients of clients and the aggregated gradients. Therefore, we can use the standard simulation proof for multi-party computation protocols to demonstrate the privacy of our generic construction. We first consider privacy protection against honest-but-curious clients who hold their own local gradients and have access to the global gradients. Specifically, let $\Pi$ denote the proposed Janus involving $n$ users $C_1, C_2, ..., C_n$ and two servers $S_0$ and $S_1$. Each user holds a local update gradient $x_i$, Janus securely computes the aggregated global update $X$. All participants may attempt to infer more additional information, the $\Pi$ satisfies the following privacy guarantee:

- For each honest-but-curious client $C_i$, the client learns nothing beyond its own local gradient $x_i$ and the final global aggregated gradient $X$. Formally, for each $C_i$, there exists a $\mathcal{PPT}$ simulator $\mathcal{S}_i$ such that:

$$\{\mathbf{View}_\Pi(C_i)\} \approx \{\mathcal{S}_i(x_i, X)\}, \tag{10}$$

  where $\mathbf{View}_\Pi(C_i)$ denotes the view of $C_i$ during the real execution of $\Pi$, $x_i$ is the $C_i$'s local updates and $X$ is the final global aggregated result.

- For $S_0$ and $S_1$, they learns nothing beyond the masked aggregated results and the aggregated results of masks. This can ensure they will learns nothing about the final global aggregated gradient $X$, thus resisting the MIA. Formally, for $S_0$ and $S_1$, there exists a $\mathcal{PPT}$ simulator $\mathcal{S}_{server}$ such that:

$$\{\mathbf{View}_\Pi(S_0, S_1)\} \approx \{\mathcal{S}_{server}(\hat{X}, CT)\}, \tag{11}$$

where $\mathbf{View}_\Pi(S_0, S_1)$ denotes the view of two servers during the real execution of $\Pi$, $\hat{X}$ is the masked aggregation result and $CT$ is the ciphertext of masks.

Given any subset $\mathcal{U} \subseteq \mathcal{C}$ of the users, where the $\mathcal{C}$ is the set of all users in the system ($|\mathcal{C}| = m$). Let the $\mathbf{REAL}_{\mathcal{U}}^{\mathcal{C},\lambda}(\{(\hat{x}_{i,t})\}_{i \in \mathcal{C}}, (c_{1,r}, c_{2,r}...c_{m,r}))$ be a random variable representing the ioint view of the users in $\mathcal{U}$. This suggests that all these honest but curious clients learned was the aggregation of gradients of all clients and their own gradients.

---

**Functionality $\mathcal{F}_s$**

Parties: users $1, \ldots, N$ from $\mathcal{S}_t$ and two servers $S_0$ and $S_1$.
Parameters: corrupted rate $\eta$, number of participating training clients per-round n.

- $\mathcal{F}_s$ receives a set of corrupted parties $\mathcal{C}$ from the adversary $\mathcal{A}$, where the $|\mathcal{C}|/|\mathcal{S}_t| \leq \eta$.

- For each round $t$:
    1. $\mathcal{F}_s$ receives a set of $N$ clients $\mathcal{S}_t$ and updates $x_{i,t}$ from clients $i \in \{\mathcal{S}_t \setminus \mathcal{C}\}$.
    2. $\mathcal{F}_s$ sends $\mathcal{S}_t$ to $\mathcal{A}$ and requests a set $M_t$. $\mathcal{F}_s$ computes the $X_t = \bigodot_{i \in \{M_t \setminus \mathcal{C}\}} x_{i,t}$ if $M_t \subseteq \mathcal{S}_t$ and continues; otherwise $\mathcal{F}_s$ sends abort to all honest participants.
    3. There are two scenarios based on whether the servers are corrupted by $\mathcal{A}$ as follows.
        - Corrupted: $\mathcal{F}_s$ outputs $X_t$ to all the participants corrupted by $\mathcal{A}$.
        - Not corrupted: $\mathcal{F}_s$ requests a mask $SK_t$ from $\mathcal{A}$ and outputs $X_t \bigodot SK_t$ to $S_1$.

---

Figure 6: Ideal functionality for Janus.

### C.3 SINGLE-ROUND SECURITY

**Theorem 1 (Security of Janus)** *Let the security parameter be $\lambda$ and $n$ be the number of users for aggregation in each round. Assuming the existence of secure underlying one-time pad, SHC, and public key encryption. Our generic construction can securely realize the ideal functionality $\mathcal{F}_s$ under the presence of a static adversary controlling $\eta$ fraction of $n$ users (and the server $S_1$) as shown in Figure 6.*

$$\mathbf{REAL}_{\Pi,\mathcal{A}}^{\mathcal{F}_s, \mathcal{F}_{sum}^t}(\lambda, n, x_{\mathcal{S}_t}) \approx \mathbf{IDEAL}_{\mathcal{F}_s, \mathcal{S}}^{\mathcal{F}_{sum}^t}(\lambda, n, x_{M_t}). \tag{12}$$

**Proof.** We first prove the security of a single round aggragation. Our generic scheme (denoted as $\Pi$) securely realizes the ideal functionality $\mathcal{F}_{sum}^t$ (Figure 7) in the random oracle model. We can find from the ideal function $\mathcal{F}_{sum}^t$ that it is the $M_t$ sent by the adversary $\mathcal{A}$ that determines the actual result. We assume the $\mathcal{A}$ controls a set of clients and denote the set of corrupted cilents as $\mathcal{C}$.

**Event 1.** We start with the servers not being corrupted by the $\mathcal{A}$. Now, we first build a simulator $\mathcal{S}$ in the ideal world, running $\mathcal{A}$ as a subroutine. Specifically, the simulation for round $t$ is as follows.

1. $\mathcal{S}$ receives a set $M_t$ from the adversary $\mathcal{A}$.

2. $\mathcal{S}$ acquires $Z_t$ from the $\mathcal{F}_{sum}^t$.

3. *Masking and Report.* $\mathcal{S}$ interacts with $\mathcal{A}$ as in the masking and report phase and acts as honest users in $i \in \{M_t \setminus \mathcal{C}\}$ with the masked updates $x'_{i,t}$ such that the $Z_t = \bigodot_{i \in \{M_t \setminus \mathcal{C}\}} x'_{i,t}$. Here the input update $x'_{i,t}$ and the mask $sk_{i,t}$ are generated by $\mathcal{S}$ itself.

4. *Collection and Aggregation.* In this phase, $\mathcal{S}$ interacts with $\mathcal{A}$, where $\mathcal{A}$ performs as a honest participant as in the collection and aggregation of $\Pi$.

5. *UnMasking and Verfication.* $\mathcal{S}$ interacts with $\mathcal{A}$ as honest participants in the unmasking and verfication phase.

6. In the above steps, if all honest participants would abort in the protocol in this round of aggregation, then $\mathcal{S}$ sends abort to $\mathcal{F}_{sum}^t$. Finally, $\mathcal{A}$ outputs the value at random and terminates this aggregation.

---

**Functionality $\mathcal{F}_{sum}^t$**

Parties: users from $\mathcal{S}_t$ and two servers.
Parameters: corrupted rate $\eta$.

- $\mathcal{F}_{sum}^t$ receives a set of corrupted participants $\mathcal{C}$ from the adversary $\mathcal{A}$ and $x_{i,t}$ from clients $i \in \{\mathcal{S}_t \setminus \mathcal{C}\}$.

- $\mathcal{F}_s$ sends $\mathcal{S}_t$ to $\mathcal{A}$ and requests a set $M_t$. $\mathcal{F}_s$ computes the $Z_t = \bigodot_{i \in \{M_t \setminus \mathcal{C}\}} x_{i,t}$ if $M_t \subseteq \mathcal{S}_t$ and continues; otherwise $\mathcal{F}_s$ sends abort to the all honest participants.

- For each round $t$:
    1. $\mathcal{F}_s$ receives a set of $N$ clients $\mathcal{S}_t$ and updates $x_{i,t}$ from clients $i \in \{\mathcal{S}_t \setminus \mathcal{C}\}$.
    2. $\mathcal{F}_s$ sends $\mathcal{S}_t$ to $\mathcal{A}$ and requests a set $M_t$. $\mathcal{F}_s$ computes the $Z_t = \bigodot_{i \in \{M_t \setminus \mathcal{C}\}} x_{i,t}$ if $M_t \subseteq \mathcal{S}_t$ and continues; otherwise $\mathcal{F}_s$ sends abort to the all honest participants.
    3. There are two scenarios based on whether the servers are corrupted by $\mathcal{A}$ as follows.
        - Corrupted: $\mathcal{F}_s$ outputs $X_t$ to all the participants corrupted by $\mathcal{A}$.
        - Not corrupted: $\mathcal{F}_s$ requests a mask $SK_t$ from $\mathcal{A}$ and outputs $Z_t \bigodot SK_t$ to $S_1$.

Figure 7: Ideal functionality for Report and Collection in Round $t$.

We construct a series of hybrid execution programs from the real world to the ideal world.

**Hybrid 1.** The view of $\mathcal{A}$ in the real-world execution is the same as the ideal world, when $\mathcal{S}$ has actual inputs from honest participants $\{x_{i,t}\}, i \in \mathcal{S}_t \setminus \mathcal{C}$, the individual masks $sk_{i,t}$ and the $SK_t$.

**Hybrid 2.** $\mathcal{S}$ does not use the actual masks in one-time pad between honest participants. It generates a random mask $sk'_{i,t}$ from the $\{0,1\}^\lambda$, then it computes the corresponding one-time pad ciphertext as $(\hat{x}'_{i,t})$. We argure the view of $\mathcal{A}$ in this hybrid is the computationally indisinguishable from the previous hybrid 1 as follows.

Firstly, the mask $sk_{i,t}$ is computed from the $\mathcal{R}_C$ of the one-time pad, and the mask $sk'_{i,t}$ is randomly sampled in the ideal world. Let the $M_t$ denotes the set of users chosen by $\mathcal{A}$ in the ideal world. $\mathcal{A}$ in the ideal and real world can observes $\mathsf{Masking}(x_{i,t}, sk_{i,t})$ between a user $i \notin M_t$ and a client $i \in M_t$. This indistinguishability stems from the selection of random masks in the specific underlying one-time pad. Secondly, $\mathcal{A}$ can observe the ciphertexts generated from the $sk'_{i,t}$. The distribution of the ciphertexts is computationally indisinguishable from the $\mathcal{A}$ observed from the real world, which is depend on the security of the underlying OTP.

**Hybrid 3.** In this hybrid, instead of using one-time pad with actural personal mask $sk_{i,t}$ randomly selected from the space $\mathcal{R}_C$, $\mathcal{S}$ uses masks randomly sampled from $\{0,1\}^\lambda$. Before the proof, we model the generation of masks as a random oracle $\mathcal{O}_R$ (see more details in the prior work (Bonawitz et al., 2017)). For $\forall i \in \{M_t \setminus \mathcal{C}\}$, the $\mathcal{S}$ samples $sk'_{i,t}$ randomly and programs $\mathcal{O}_R$ as $sk'_{i,t} = \hat{x}_{i,t} \oslash x_{i,t}$, where the $\hat{x}_{i,t}$ is observed in the real world and the $\oslash$ denotes the inverse operation of $\odot$. From the perspective of $\mathcal{A}$, the distributions of $\hat{x}_{i,t}$ in this hybrid and the previous one are statistically indistinguishable.

Additionally, $\mathcal{A}$ learns the $sk'_{i,t}$ in the clear for $i \in M_t$ in the real and ideal world. The distributions of $sk'_{i,t}$ are identical. However, $\mathcal{A}$ learn nothing about $sk'_{i,t}$ for $i \notin M_t$ in both worlds because of the semantic security of the underlying one-time pad. From the view of $\mathcal{A}$, this hybrid is computationally indistinguishable from the previous hybrid.

**Hybrid 4.** In this hybrid, instead of control the random oracle as in the previous hybrid, $\mathcal{S}$ will program the random oracle $\mathcal{O}_R$ as $sk'_{i,t} = \hat{x} \oslash x'_{i,t}$. Specifically, the $x'_{i,t}s$ are chosen such that $\bigodot_{i \in \{M_t \setminus \mathcal{C}\}} x_{i,t} = \bigodot_{i \in \{M_t \setminus \mathcal{C}\}} x'_{i,t}$. From the view of $\mathcal{A}$ this hybrid is the same as the previous one, which can be derived from Lemma 6.1 of the prior work (Bonawitz et al., 2017).

**Hybrid 5.** Similar to the previous operation, this hybrid replaces the mask of honest participants with the result from the random oracle. $\mathcal{S}$ will abort if the $\mathcal{A}$ would cheat by sending invalid masked updates to $\mathcal{S}$. In the phase of unmasking and verification, the $\mathcal{A}$ would cheat by sending different $M_t$

to $\mathcal{F}^t_{sum}$. $\mathcal{S}$ will simulate the following protocol (see as Lemma 1) and output whatever the protocol outputs. It is identical to the previous hybrid by doing this.

The final hybrid precisely represents the execution of the ideal world. The aforementioned events indicate that our system is secure in the ideal world with a single round process.

**Event 2.** In this event, the server is not corrupted by $\mathcal{A}$, the whole simulation is the same as Event 1, except that the $\mathcal{S}$ will program the masks added by the $\mathcal{A}$ in each step.

We complete the proof that for any single round $t$, the protocol $\Pi$ always securely realizes the ideal functionality $\mathcal{F}^t_{sum}$ in the presence of a static malicious adversary.

**Lemma 1** *Assume there exists a PKI and a secure signature scheme, there are $3\zeta$ particpants with at most $\zeta$ colluding malicious participants. Specifically, each party has an input bit of $0$ or $1$ from a server. There exists a one-round protocol enabling each honest participant to determine whether the server sent the same value to all honest participants.*

**Proof.** When an honest participant receives at least $2\zeta$ messages with the same value, it indicates that the server has sent the same value to all honest participants. This is because, in the given system, the threshold of $2\zeta$ identical messages can only be met if a large majority of honest participants have received the same value. Specifically, let the total number of participants in the system be $n = t_h + t_m$, where $t_h$ denotes the number of honest participants and $t_m$ denotes the number of malicious participants. For security and consistency in distributed protocols, the parameter $\zeta$ is set such that $t_h > 2\zeta$. When an honest participant receives no fewer than $2\zeta$ identical messages, it can confidently conclude that at least $\zeta + 1$ of these messages were sent by distinct honest participants, ensuring consistency of the message content. Hence, it can be inferred that the server has broadcast the same value to all honest participants.

Conversely, if an honest participant receives fewer than $2\zeta$ messages with the same value, this suggests that the server may have sent different messages to different participants during the communication process. Since the number of identical messages received by honest participants falls short of forming a consensus of $2\zeta$, it implies that the server may have engaged in malicious behavior by sending inconsistent messages to various honest participants. To ensure the security and consistency of the protocol, in such a scenario, the honest participant will abort the protocol execution. This abort mechanism effectively prevents potential security threats and data integrity issues that could arise due to inconsistent messages from the server.

## C.4 MULTI-ROUND SECURITY

Our threat model assumes the corrupted rate is $\eta$, which means that $\mathcal{A}$ controls $\eta n$ clients throughout total $T$ rounds. In order to prove the security of the multi-round scheme on the basis of the above single-round security proof. The mask $sk_{i,t}$ computed from $\mathcal{O}_R$ of the underlying SHC $\Pi_\mathcal{C}$. Let the $\Delta_t$ be distribution of the view of $\mathcal{A}$ in the single round $t$ and the total number of rounds needed for the model to converge is $T$. If there exists an adversary $\mathcal{B}$, and two rounds of aggregation $t_1, t_2 \in [0, T]$, where $\mathcal{B}$ can distinguish between $\Delta_{t_1}$ and $\Delta_{t_2}$, then we can construct an adversary $\mathcal{A}$ breaks the security of the underlying $\Pi_\mathcal{C}$. We call the challenger in the security game of $\Pi_\mathcal{C}$ as $\mathcal{S}$. Specifically, there exists two worlds ($b = 0$ or $1$) for the $\mathcal{O}_R$ game. The $\mathcal{S}$ uses a random function if the $b = 0$. When $b = 1$, $\mathcal{S}$ actual $\Pi_\mathcal{C}$. Then we build the $\mathcal{A}$ as follows. On input $t_1, t_2$ from $\mathcal{B}$, the $\mathcal{A}$ asks for $sk_{i,t}$ for all honest participants in the round $t_1$ and $t_2$. Then $\mathcal{A}$ could computes the masked updates from the $\Pi$ prescribed. It generates two views $\Delta_{t_1}, \Delta_{t_2}$ and sends them to the $\mathcal{B}$. Finally, $\mathcal{A}$ outputs whatever the $\mathcal{B}$ outputs as the answer.

## C.5 RESISTING MIA

The MIA is effective primarily because the server is aware of the final aggregated result. If the server can manipulate the parameters sent to different clients, it can introduce inconsistencies that influence the model training process. The key to resisting this attack is to ensure that all clients start with the same initial model parameters. This uniformity can be achieved through two main approaches: using a public bulletin board where the initial parameters are posted for everyone to see, or through mutual agreement among clients to verify that the parameters they receive are indeed consistent

across the network. The public bulletin board approach suffers from centralized dependency, information leakage, and scalability issues, while the mutual agreement method has high communication complexity, scalability limitations, and is vulnerable to Sybil attacks. Both methods face challenges in maintaining consistency and security as the number of clients increases.

A significant advancement in Janus, which brought forward a novel concept: making the aggregation results in the system visible only to the clients. In this approach, the computation of the final aggregation result is performed locally by each client, rather than centrally by the server. This means that even if the server, denoted as $S_0$, disseminates inconsistent model parameters to different clients, it remains unaware of the actual final aggregated model. This paradigm shift ensures that the server cannot gain insight into the final result, thus preventing it from introducing systematic biases.

Additionally, we assumes that $S_0$ and the clients are not colluding. In other words, the server cannot conspire with any client to manipulate the aggregation process. By decentralizing the aggregation computation and keeping the final result private among the clients, the Janus effectively mitigates the risk of a successful MIA. This approach not only enhances the security of the federated learning framework but also reinforces the privacy and trustworthiness of the system by limiting the server's influence over the final model.

# D  APPENDIX FOR REBUTTAL REVISION

## D.1  MORE COMPARISON METHODS

Table 3 demonstrates that Janus surpasses other state-of-the-art schemes in terms of security, efficiency, and functionality. Specifically, our scheme achieves optimal efficiency while providing enhanced security and functionality. Our scheme makes weaker assumptions Compared to ELSA Rathee et al. (2023), resulting in higher security while supporting multi-round aggregation with a significant performance improvement. Compared to VeriFL Guo et al. (2021), Janus does not require constructing complex communication graphs or performing time-consuming secret sharing operations, which leads to substantial performance gains.

Table 3: Comparison of SA Constructions

| Scheme | Input Privacy | Multi-round | Verifiability | Dynamic | Versatility | NS* | Efficience‡ | MIA |
|---|---|---|---|---|---|---|---|---|
| SecAgg (Bonawitz et al., 2017) | ✓ | ✗ | ✗ | ✗ | ✗ | 1 | ◯ | ✗ |
| BBSA (Bell et al., 2020) | ✓ | ✗ | ✗ | ✗ | ✗ | 1 | ◕ | ✗ |
| VeriFL Guo et al. (2021) | ✓ | ✗ | ✓ | ✗ | ✗ | 1 | ◔ | ✗ |
| ELSA Rathee et al. (2023) | ✓ | ✗ | ✗ | ✗ | ✗ | 2 | ◕ | ✗ |
| Flamingo (Ma et al., 2023) | ✓ | ✓ | ✗ | ✗ | ✗ | 2† | ◕ | ✗ |
| Janus | ✓ | ✓ | ✓ | ✓ | ✓ | 2 | ● | ✓ |

✓ Support, ✗ No support. Versatility: A generic construction. ⋆ Number of servers. † The decryptors of this construction can be abstracted to a server. ‡ More black parts in the circle indicate better efficiency, and the theoretical support comes from the computation efficiency analysis in Table 2.

## D.2  ADDITIONAL EXPERIMENTS

Our scheme is not limited to specific models or datasets. To better support this conclusion, we have added more experimental results for the CIFAR-100 dataset on different models. Specifically, the CIFAR-100 dataset is a challenging benchmark dataset widely used in machine learning and computer vision research. It contains 60,000 color images, each of size 32x32 pixels, distributed across 100 distinct classes. Each class is organized hierarchically, with 20 superclasses grouping the 100 fine-grained categories, adding a layer of complexity. This fine-grained nature, combined with the small image resolution, makes classification tasks on CIFAR-100 particularly difficult, as it demands models to capture subtle features and patterns. The dataset is balanced, with each class containing 500 training images and 100 test images, ensuring uniform representation while amplifying the difficulty of distinguishing between visually similar categories. The specific experimental results are shown in Figure 8 and 9, which show that Janus is comparable to most of the existing schemes in terms of performance, but Janus has an obvious advantage in terms of computational overhead.

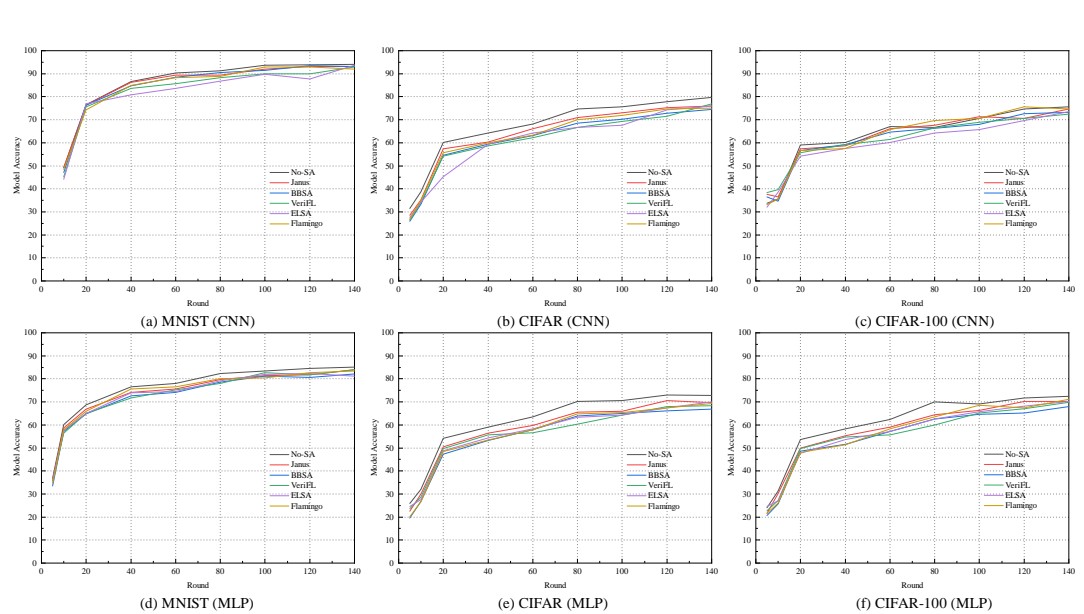

Figure 8: Test accuracy across different datasets and models.

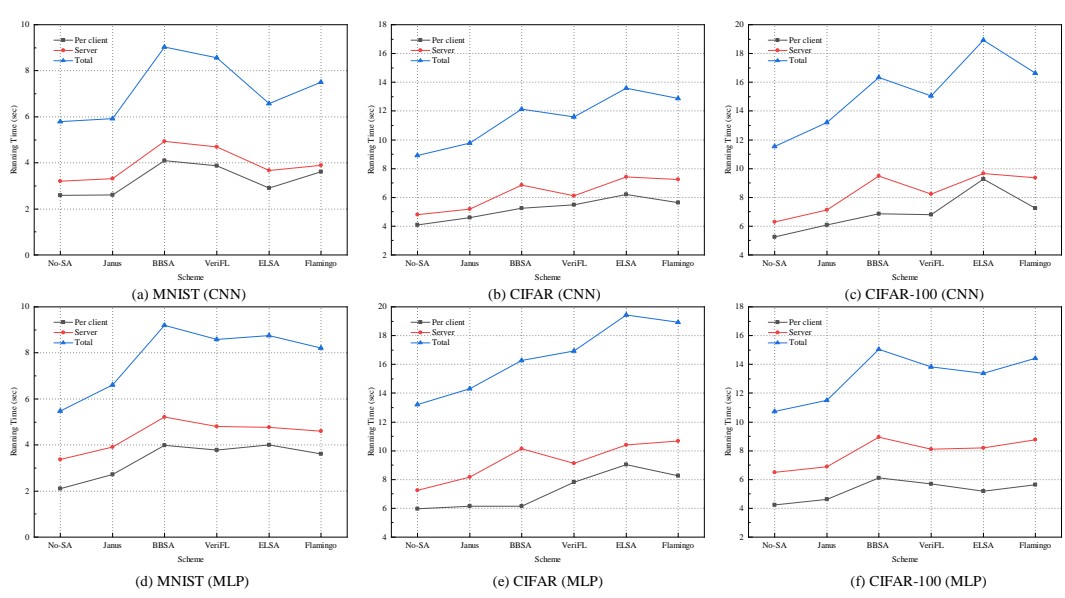

Figure 9: Computation overhead across different datasets and models.

