# OpenReview forum: "Janus: Dual-server Multi-Round Secure Aggregation with Verifiability for Federated Learning"
_ICLR.cc/2025/Conference — Submitted to ICLR 2025_

### Official Review · Reviewer_Xv5P · 2024-10-30

**Soundness:** 1
**Presentation:** 2
**Contribution:** 1
**Rating:** 3
**Confidence:** 5

**Summary:**

Janus introduces a Secure Aggregation (SA) scheme for Federated Learning (FL) that overcomes some challenges in existing protocols by implementing a dual-server architecture and a  cryptographic primitive called Separable Homomorphic Commitment (SHC).

**Strengths:**

The paper is clear and well-organized, with complex concepts explained effectively and supported by useful visual aids.

**Weaknesses:**

While the paper introduces an approach with Janus, several weaknesses limit its contribution and practical applicability. First, it lacks a comparison with single-server state-of-the-art methods like VeriFL, raising questions about its feasibility and efficiency compared to existing solutions. The assumption that clients do not collude is unrealistic, especially if clients are interested in each other's model updates; the scheme does not specify how many colluding users it can tolerate, leaving potential vulnerabilities unaddressed. Fault tolerance is insufficient, as user dropout can disrupt the entire aggregation process, and the paper does not explain how it handles partial user failures or higher dropout rates. The scheme may be vulnerable to differential attacks if an attacker obtains masked data from multiple rounds and exploits similarities in user inputs to infer private information. The commitment mechanism lacks detailed specifications, and if a simple hash-based commitment is used, it may be susceptible to length extension attacks. The Separable Homomorphic Commitment (SHC) appears to be a variant of existing commitment schemes without substantial innovation and lacks essential properties like trapdoor mechanisms and equivalence, potentially weakening security; more theoretical support and security proofs are needed. Additionally, implementation details for comparative schemes are insufficient, experiments lack statistical significance analysis and detailed breakdowns of computational and communication overhead, and there is no evaluation of scalability with different user numbers or model sizes. Claims of resistance to Model Inconsistency Attacks and multi-round security are not experimentally validated, which undermines the credibility of the proposed security enhancements.

**Questions:**

1. The paper does not compare Janus's overhead and feasibility with existing single-server SOTA methods that achieve similar privacy and verifiability. Notably, schemes like VeriFL have demonstrated efficient verifiable federated learning in a single-server setting. In addition, the aggregation results of a single server can actually be kept secret (already exists).

2. The paper assumes that clients do not collude. However, if clients are interested in each other's model updates, this assumption may not hold. Colluding clients could potentially infer private information about other users.

3. The scheme's fault tolerance is limited; user dropout can adversely affect the entire aggregation process. The paper does not adequately explain how the system handles partial user failures or higher dropout rates.

4. Develop and describe mechanisms to handle user dropout more effectively. This could include techniques like dropout resilience protocols or asynchronous aggregation methods. Experiment with varying dropout rates, including those higher than the idealistic 10%, to demonstrate the scheme's robustness in realistic settings.

5. If an attacker obtains masked inputs from two rounds where the user's input remains similar (i.e., \( x_{i,t} \approx x_{i,t+1} \)), they could perform differential analysis to infer changes in the original inputs.

6. The paper does not specify the specific requirements or properties of the commitment algorithm used in the SHC. If a simple hash-based commitment is used (e.g., \( c_{i,t} = H(x_{i,t} || r_{i,t}) \)), it may be vulnerable to length extension attacks or other cryptographic weaknesses.

7. SHC is described as a variant of existing commitment schemes but lacks substantive innovation. It does not support essential properties like trapdoor mechanisms or equivalence, which are present in schemes like the one used in VeriFL. The security proofs provided are insufficient to establish its robustness.

8. The implementation details, particularly the parameter settings for Janus and the comparative schemes (BBSA and Flamingo), are not thoroughly documented. This omission hampers reproducibility and makes it difficult to assess the validity of the experimental results.

9. The experiments lack information on the number of repetitions and do not include statistical analyses to determine the significance of the results.

10. The paper only reports the total computation time without decomposing the overhead into its constituent components (e.g., cryptographic operations, communication latency).

11. There is no empirical data or graphical analysis of the communication overhead; the paper relies solely on theoretical analysis.

12. The experiments are conducted with a relatively small number of users (100), which is insufficient to demonstrate scalability. The impact of varying the number of users and the model size on performance is not evaluated.

13. While the paper claims that Janus resists MIA, it does not provide experimental tests or simulations to substantiate this claim.

14. The security of Janus over multiple rounds is not experimentally verified, leaving potential vulnerabilities unaddressed.

15. The scheme does not specify the maximum number of colluding users it can tolerate without compromising security.

16. By omitting features like trapdoor mechanisms and equivalence properties, SHC may be less secure than existing commitment schemes. For example, in VeriFL, the commitment scheme allows for equivalence operations, which enhance functionality and security.

---

> ### Author Response · Authors · 2024-11-14
>
> Thank you for your valuable feedback, and we address your concerns as follows.
>
> $\textbf{1. Security (Q1, Q2, Q5, Q13-15) }$.
>
> Q1: Unlike existing single-server schemes (e.g., VeriFL), which require the server to aggregate results and thus fail to resist MIA, our Janus scheme enables dynamic user participation while resisting MIA and ensuring verifiability. Additionally, Janus is more efficient, relying on simple one-time pads (OTP) and SHC, rather than complex cryptographic tools like homomorphic encryption or secret sharing. We will highlight these advantages in the revised version.
>
> Q2: Janus resists collusion of up to $n-2$ clients. If $n-1$ clients collude, they can trivially deduce the remaining client’s information. However, with $n-2$colluding clients, they can only obtain the additive result of two uncolluding clients. This result is an aggregation of two encrypted or obfuscated values, making it impossible to recover each uncolluding user's specific gradient information. We will provide a detailed analysis of this property for clarity.
>
> Q5: The masking technique used in $\textit{Janus}$ is OTP. As long as each secret key ($sk$) is used only once per round, the masked inputs reveal no information about the original values, even a single bit. Therefore, differential analysis is ineffective in this context.
>
> Q13–15: Our theoretical analysis supports the claimed properties of resisting MIA, supporting multiple rounds, and tolerating a maximum number of colluding clients. This is similar to the approaches used in Flamingo (SP’23) and Securing Secure Aggregation (AAAI’23), where theoretical analysis is also employed.
>
> $\textbf{2. SHC (Q6-7, Q16)}$.
>
> In Section 3.1, we clarify that SHC requires completeness, binding, hiding, separability, and homomorphism. The hash-based commitment you mentioned does not meet these requirements, so length extension attacks do not apply. Our design does not require additional properties like trapdoors or equivalence (as in VeriFL), which simplifies the design and improves performance without compromising security. The Pedersen commitment, a widely used SHC, serves as a strong example of our approach’s security, as demonstrated in blockchain systems like Monero and ZCash.
>
> $\textbf{3. Dropout (Q3, Q4)}$.
>
> Q3: In traditional schemes, all users need to negotiate a shared key as a mask before communication. If a user fails to upload masked parameters in later stages, the mask cannot be canceled across the system. However, our scheme avoids establishing complex communication graphs and only requires a single interaction with the server, thus eliminating the dropout issues present in traditional schemes. The only potential problem arises if a user fails to update both servers, but this is trivially avoided by ensuring synchronization between the two servers. This ensures that both servers either receive the message or don’t receive it at all, thus maintaining atomicity. However, traditional schemes cannot resolve dropout issues via this simply synchronization method.
>
> Q4: We will include experiments with varying dropout rates in the revised version. These do not involve complex techniques and will help validate our claims.
>
> $\textbf{4. Experiments (Q8-Q12)}$.
>
> In lines 438-443, we have explained the specific parameters used in the experimental setup to ensure reproducibility. Our scheme is not limited to specific models or datasets, and the paper includes theoretical analysis and experimental comparisons of each scheme’s performance across different models and datasets (lines 362-564). The theoretical analysis shows that our scheme significantly improves computational efficiency. To further highlight the advantages of $\textit{Janus}$, we will add experiments validating performance in terms of computational and communication costs. We have already conducted multi-round aggregation experiments, which are similar to those with varying numbers of users. Additionally, we will include new datasets (e.g., CIFAR-100, Fashion-MNIST) and advanced comparison schemes (e.g., Elsa (SP’23), VeriFL (2020)) in the updated version.
>
> We greatly appreciate your feedback and will ensure these clarifications are included in the revised manuscript. If you have any further concerns, please let us know.

---

> > ### Author Response · Authors · 2024-11-20
> > **Revised paper uploaded**
> >
> > Thank you again for your time and valuable feedbacks on our work. We have carefully addressed your concerns in the revised paper we have now submitted. Specifically, we have made the following updates to address your concerns.
> >
> > $\textbf{Uncolluding Assumptions:}$ We have revised the uncolluding assumptions, which are now included in Section C.1 for clarity and comprehensiveness. This addresses your concerns about security issues such as collusion, Sybil attacks, and other risks.
> >
> > $\textbf{More SOTA Comparison Schemes:} $ We have added two new SOTA schemes in Section D.1 to analyze and compare them theoretically. Specifically, we incorporate two new SOTA comparison methods, VeriFL and ELSA. We provide both theoretical and experimental analyses to highlight their implications and comparisons.
> >
> > $\textbf{Additional Experiments:} $ Our scheme is not limited to specific models or datasets. To better support this conclusion, we have added more SOTA methods and experiments as your suggestions.
> > 1. Sections D.1 and D.2 now incorporate two new SOTA methods, VeriFL and ELSA as your recommendation. We provide both theoretical and experimental analyses to highlight their implications and comparisons.
> > 2. In Section D.2, we also add more experimental results using the more complex CIFAR-100 dataset across different models to further validate our conclusions.
> >
> > We believe these revisions address your concerns and strengthen the paper. We would appreciate your reevaluation of the updated version. Please feel free to reach out if you have any further questions, and we look forward to your feedback.

---

### Official Review · Reviewer_sXBs · 2024-10-31

**Soundness:** 2
**Presentation:** 2
**Contribution:** 2
**Rating:** 3
**Confidence:** 5

**Summary:**

The paper introduces Janus, a system for multi-round secure aggregation for federated learning. By having a very low round setup independent of the number of clients, Janus can easily be used for multiple rounds. Janus utilizes a dual-server setup where one server handles masked updates and the other manages aggregation of masks, ensuring that neither server has access to the final aggregated results. A novel cryptographic primitive, Separable Homomorphic Commitment, enables client-side verification of the correctness of the aggregation result. The authors evaluate Janus end-to-end, highlighting its constant per-client communication and computational overhead while preserving model accuracy.

**Strengths:**

- Addresses scalability challenges of privacy-preserving federated learning
- Comprehensive evaluation with end-to-end model training

**Weaknesses:**

- Incomplete comparison with related work in the multi-server FL setting such as 2-party MPC or [ELSA: Secure Aggregation for Federated Learning with Malicious Actors, S&P’23]. These approaches also provide low overhead for clients, independent of the number of clients. Adding a more detailed comparison could help contextualize Janus' unique contributions and highlight differences in scalability and overhead reduction strategies.
- The threat model assumes non-collusion among entities, which may not align with practical scenarios, particularly regarding client behaviors. In real-world applications, service providers could potentially collude with a bounded subset of clients, as assumed in related works’ threat models. Janus’ reliance on a non-collusion model raises concerns about its susceptibility to model inconsistency attacks if a provider colludes with even a single client or introduces a Sybil client, potentially gaining access to the aggregated model and undermining security. Additionally, the threat model assumes the servers are semi-honest, which makes achieving verifiability property trivial.
- The security proof contains inaccuracies that hinder their clarity. For instance, Theorem 1 references two distinct ideal functionalities (Figures 6 & 7), though Figure 7 appears underdefined. The distinction in scenarios based on “whether the servers are corrupted by A” might incorrectly imply that both servers could be corrupted simultaneously, which conflicts with the intended security assumptions.

**Questions:**

1. In Table 1: What is the versatility property?
2. How does your approach compare with a straightforward 2PC baseline, and other multi-server FL systems such as [ELSA: Secure Aggregation for Federated Learning with Malicious Actors, S&P’23]?
3. Given that you focus much of the evaluation on the accuracy and loss of the approaches, would we expect a difference with related secure aggregation schemes?
4. How can new users join the training process? What prevents the service provider from setting up sybil clients to get access to the model?

---

> ### Author Response · Authors · 2024-11-14
>
> Thank you for your insightful feedback. Below, we address your concerns.
> 1. $ \textbf{Versatility}$. This property highlights that our scheme is a generic construction, not limited to specific cryptographic tools like Separable Homomorphic Commitments (SHC) or one-time pads (OTP). In contrast, other schemes rely on specific tools, lacking such flexibility. We will add a description of this versatility in the revised paper.
>
> 2. $\textbf{Comparison with 2PC and Other Schemes}$. While 2PC is simple and direct, it often relies on resource-intensive homomorphic encryption and zero-knowledge proofs. In contrast, our scheme uses lightweight cryptographic primitives (SHC and OTP). ELSA assumes at least one honest server, whereas our scheme only requires two uncolluding servers, allowing for malicious server aggregation. We will correct the threat model in the revised paper to reflect that servers can be malicious. Our scheme can detect malicious aggregation through its built-in verifiability, which provides an advantage over these approaches.
>
> 3. $\textbf{Advantages}$. In addition to comparing accuracy and loss, we evaluate the single-round computational cost of each scheme across different models and datasets (lines 483–524). Our analysis shows that while secure aggregation inevitably increases computational cost compared to plaintext aggregation, our scheme achieves substantial efficiency improvements over other advanced secure aggregation schemes of the same type.
>
> 4. $\textbf{User Join and Sybil Attacks Resistance}$. In lines 23–25 and lines 301-308, we explain how new users can join the training process. A new user simply needs to acquire the system’s public parameters, generate a public-private key pair, and obtain the server’s public key. The key is certified by authorities to authenticate the user’s identity. This approach avoids the need for rebuilding the communication graph when users leave, which distinguishes our scheme from others. An adversary attempting a Sybil attack could obtain encrypted data, but would not be able to perform a MIA, as controlling at least $n-1$ users would be required to access private input. Additionally, our scheme’s reliance on Public Key Infrastructure (PKI) and the use of certificates makes Sybil attacks even more difficult, as forgeries would require creating valid certificates from authorities. Thus, our scheme effectively mitigates Sybil attack risks.
>
> We greatly appreciate your feedback and will incorporate these clarifications in the revised paper. Thank you again, and we look forward to your response.

---

> > ### Author Response · Authors · 2024-11-20
> > **Revised paper uploaded**
> >
> > Thank you again for taking the time to provide valuable comments on our work. We have carefully responded to your concerns in the revised paper we have now submitted. Specifically, we have made the following updates to address your concerns.
> >
> > $\textbf{Uncolluding Assumptions:}$ We have revised the uncolluding assumptions, which are now included in Section C.1 for clarity and comprehensiveness. This addresses your concerns about security issues such as collusion, Sybil attacks, and other risks.
> >
> > $\textbf{More SOTA Comparison Schemes:}$ We have added two new SOTA schemes in Section D.1 to analyze and compare them theoretically. Specifically, we incorporate two new SOTA comparison methods, VeriFL and ELSA. We provide both theoretical and experimental analyses to highlight their implications and comparisons.
> >
> > We believe the revisions fully address your concerns and strengthen the paper. We would greatly appreciate your reevaluation and are happy to provide any further clarification if needed. We look forward to your feedback.

---

> > > ### Comment · Reviewer_sXBs · 2024-11-26
> > >
> > > Thank you for your detailed response. While I appreciate your effort to address my concerns, there are still aspects that remain unresolved:
> > >
> > > 1. **Versatility.** You describe your scheme as more “versatile” compared to others, but it remains unclear what specific non-blackbox constructions other schemes rely on that reduce their versatility. Could you clarify and provide examples or references to substantiate this claim?
> > > 2. **Comparison with ELSA**. Thank you for including the comparison. You report a theoretical efficiency metric, but the basis for these values is unclear. Could you elaborate on how these results were derived? Additionally, since you provide model accuracy for end-to-end training with ELSA, it would strengthen your evaluation if you could include empirical runtime performance results for the comparison.
> > > 3. **Reporting of accuracy and loss.** You did not address my question about why so much space is used in the paper to report accuracy and loss values. “Given that you focus much of the evaluation on the accuracy and loss of the approaches, would we expect a difference with related secure aggregation schemes?”
> > > 4. **Model Inconsistency Attacks.** In your response, you state: *"An adversary attempting a Sybil attack could obtain encrypted data, but would not be able to perform a Model Inconsistency Attack (MIA), as controlling at least n−1n-1n−1 users would be required to access private input."* However, wouldn’t it be sufficient for the server to collude with just one client, since clients receive model outputs in plaintext? If so, this seems highly unrealistic given the feasibility of Sybil attacks. While the non-collusion assumption between the two servers could be plausible if you provide concrete examples of feasible settings, the assumption of non-collusion between the server and any client seems overly strong.

---

> > > > ### Author Response · Authors · 2024-11-26
> > > > **Response to Reviewer sXBs - part 1/2**
> > > >
> > > > Dear Reviewer sXBs,
> > > >
> > > > Thank you for the opportunity to discuss our paper further. We would like to address your concerns as follows.
> > > >
> > > > 1. Versatility. You describe your scheme as more “versatile” compared to others, but it remains unclear what specific non-blackbox constructions other schemes rely on that reduce their versatility. Could you clarify and provide examples or references to substantiate this claim?
> > > >
> > > > $\textbf{Response 1:}$ In our paper, the SHC is a blackbox component. Versatility means that all SHC schemes can be used to construct an instantiation scheme, such as Pedersen commitment, ElGamal-based commitment, etc. Thus, our Janus is a generic construction. However, to the best of our knowledge, other SOTAs just give a single specific scheme from double-masking e.g., [1], [2], [3], which is a specific non-blackbox construction.
> > > >
> > > > [1] Guo X, Liu Z, Li J, et al. Verifl: Communication-efficient and fast verifiable aggregation for federated learning[J]. IEEE Transactions on Information Forensics and Security, 2020, 16: 1736-1751.
> > > >
> > > > [2] Ma Y, Woods J, Angel S, et al. Flamingo: Multi-round single-server secure aggregation with applications to private federated learning[C]//2023 IEEE Symposium on Security and Privacy (SP). IEEE, 2023: 477-496.
> > > >
> > > > [3] Bonawitz K, Ivanov V, Kreuter B, et al. Practical secure aggregation for privacy-preserving machine learning[C]//proceedings of the 2017 ACM SIGSAC Conference on Computer and Communications Security. 2017: 1175-1191.
> > > >
> > > >
> > > > 2. Comparison with ELSA. Thank you for including the comparison. You report a theoretical efficiency metric, but the basis for these values is unclear. Could you elaborate on how these results were derived? Additionally, since you provide model accuracy for end-to-end training with ELSA, it would strengthen your evaluation if you could include empirical runtime performance results for the comparison.
> > > >
> > > > $\textbf{Response 2:}$ We have added the theoretical sources for the efficiency evaluation in Table 2 of Section 4.1. Specifically, we have updated the computational overhead complexity analysis of each scheme in Table 2 on page 8. We have also provided the empirical runtime to support the theoretical analysis. Please refer to Figure 9 for the empirical runtime results.
> > > >
> > > > 3. Reporting of accuracy and loss. You did not address my question about why so much space is used in the paper to report accuracy and loss values. “Given that you focus much of the evaluation on the accuracy and loss of the approaches, would we expect a difference with related secure aggregation schemes?”
> > > >
> > > > $\textbf{Response 3:}$ Our experiments focus on verifying the impact of introducing secure aggregation on the model performance from the perspectives of model accuracy and loss. These are also mainly discussed in the existing SOTAs. From the results, we demonstrate that like existing SOTAs, our proposal can ensure the privacy protection, with almost no compromise in model accuracy. We also compare the running time among our proposal and existing SOTAs. The experimental results show that the system overhead of our scheme has been significantly reduced (Table 2, Figure 9) with acceptable model loss performance (Figure 8).

---

> > > > > ### Author Response · Authors · 2024-11-26
> > > > > **Response to Reviewer sXBs - part 2/2**
> > > > >
> > > > > 4. Model Inconsistency Attacks. In your response, you state: "An adversary attempting a Sybil attack could obtain encrypted data, but would not be able to perform a Model Inconsistency Attack (MIA), as controlling at least $n-1$ users would be required to access private input." However, wouldn’t it be sufficient for the server to collude with just one client, since clients receive model outputs in plaintext? If so, this seems highly unrealistic given the feasibility of Sybil attacks. While the non-collusion assumption between the two servers could be plausible if you provide concrete examples of feasible settings, the assumption of non-collusion between the server and any client seems overly strong.
> > > > >
> > > > > $\textbf{Response 4:}$ We are not assuming “non-collusion between the server and any client”, whereas we assume “server can successfully perform a MIA, as controlling at least $n-1$ clients”. Maybe the claim in the abstract—“Our dual-server model separates aggregation tasks, ensuring that neither server has access to the final aggregated results, thus effectively preventing MIA”—causes confusion. You might interpret it as implying that once a server obtains the final aggregated result, it can successfully launch a MIA. We have corrected the sentence to “Our dual-server model separates aggregation tasks, ensuring that neither server can successfully launch a MIA without controlling at least $n-1$ clients”.
> > > > > More specifially, if the server colludes with one client, it can indeed obtain the final plaintext model output. However, in this situation, MIA (lines 691-701) can only be successfully initiated when the entire system contains only 2 clients. When the system contains $n>=3$ clients, the server colludes with one clients who can obtain the model output but cannot successfully initiate an MIA. For ease of understanding, let's assume the system has four clients, A,B,C,D ($n=4$) and a server, S. The successful MIA is as follows, S wants to obtain the parameters of A. S colludes with B,C,D (controlling $n-1=3$ clients). S can distribute crafted initial parameters to B, C, and D. This can trigger $\textit{dying-ReLU}$ and make the inputs of B,C,and D be zero. The final aggregation result is three zeros plus the actual inputs of A. Thus, S can successfully perform a MIA to get the input of A. However, if S controls only 2 or only 1 non-target clients (controlling clients<n-1), then S can only get the sum of the inputs from A and other clients who are not controlled. In practical application scenarios, $n$ is usually a very large number, and it will cause high costs or be at a lost for S to control at least $n-1$ clients. In addition, [1], [2], [3] use secret sharing technology, thus usually assume that at least $n-1$ clients collude to make the system insecure as our assumption. Finally, the assumption of two uncolluding servers is common and reasonable in this area, as demonstrated by works like [3], [4] and [5], all of which make similar assumptions. We have illustrated the practical applicability of this assumption (lines 77-81).
> > > > >
> > > > > [1] Bell J H, Bonawitz K A, Gascón A, et al. Secure single-server aggregation with (poly) logarithmic overhead[C]//Proceedings of the 2020 ACM SIGSAC Conference on Computer and Communications Security. 2020: 1253-1269.
> > > > >
> > > > > [2] Bonawitz K, Ivanov V, Kreuter B, et al. Practical secure aggregation for privacy-preserving machine learning[C]//proceedings of the 2017 ACM SIGSAC Conference on Computer and Communications Security. 2017: 1175-1191.
> > > > >
> > > > > [3] Ma Y, Woods J, Angel S, et al. Flamingo: Multi-round single-server secure aggregation with applications to private federated learning[C]//2023 IEEE Symposium on Security and Privacy (SP). IEEE, 2023: 477-496.
> > > > >
> > > > > [4] Guo X, Liu Z, Li J, et al. Verifl: Communication-efficient and fast verifiable aggregation for federated learning[J]. IEEE Transactions on Information Forensics and Security, 2020, 16: 1736-1751.
> > > > >
> > > > > [5] Rathee M, Shen C, Wagh S, et al. Elsa: Secure aggregation for federated learning with malicious actors[C]//2023 IEEE Symposium on Security and Privacy (SP). IEEE, 2023: 1961-1979.
> > > > >
> > > > >
> > > > > Thanks again for your efforts and time. We would greatly appreciate it if you could review the updated paper and let us know if it addresses your concerns.
> > > > >
> > > > > Best regards,
> > > > >
> > > > > Authors

---

### Official Review · Reviewer_3v49 · 2024-11-03

**Soundness:** 3
**Presentation:** 3
**Contribution:** 3
**Rating:** 6
**Confidence:** 4

**Summary:**

The paper presents Janus, a privacy-enhanced multi-round secure aggregation (SA) scheme for federated learning. It addresses challenges faced by existing protocols like Flamingo, including dynamic user participation, model inconsistency attacks (MIA), and lack of verifiability. Janus uses a dual-server architecture and a new cryptographic primitive, Separable Homomorphic Commitment (SHC). New users can easily join training, and the dual-server setup prevents MIA. SHC ensures aggregation result verifiability. Experiments show improved security and efficiency with reduced per-client overhead and maintained model accuracy.

**Strengths:**

(1) The dual-server architecture and the concept of separable homomorphic commitmen are novel contributions. The combination of these elements to address multiple challenges in secure aggregation is interesting.

(2) The scheme is well-designed, with each component serving a specific purpose in enhancing security and efficiency. The integration of SHC with the dual-server model is seamless and effective.

**Weaknesses:**

(1) While the non-collusion assumption of servers is stated, a more in-depth analysis of potential threats and how they are mitigated in different scenarios could be added. For example, what if a malicious actor compromises one of the servers or if there are side-channel attacks?

(2) The experiments could be more extensive. For instance, testing on a wider range of datasets and models, including those with more complex architectures and larger data volumes, would provide a more comprehensive evaluation of the scheme's performance. Also, the impact of different network conditions on the performance of Janus could be explored.

(3) Although the author has compared with some existing methods, there are few comparison methods. The author may need to add some comparison methods to further verify the effectiveness of Janus.

(4) The author may also need to add some additional experiments to verify the effectiveness of Janus, such as aggregation completion time, computation costs, etc.

**Questions:**

The author needs to explain in detail the questions I mentioned above, and I will determine the final score based on your answers.

---

> ### Author Response · Authors · 2024-11-14
>
> We sincerely thank you for your time and valuable feedback. Below, we address the technical and theoretical concerns you raised.
> 1. $\textbf{Security Assumption}$. The specific assumptions in our paper are as follows: The two servers will not collude but may perform incorrect aggregation. The scheme also allows for up to $n-2$ clients to collude. Specifically, even if the server aggregates incorrect results, our scheme provides verifiability, which enables us to detect such behavior and mitigate the associated risks.
> If the server colludes with up to $n-2$ clients, it can only obtain the additive result of the remaining two uncolluding clients. This result is an aggregation of two encrypted or obfuscated values, making it impossible to recover each uncolluding user's specific gradient information. This ensures that the colluding entities cannot initiate a Model Inconsistency Attacks (MIA) or access the private information of the remaining two non-colluding clients. When $n-2$ clients collude, this assumption is even weaker, as the absence of server involvement further limits the accessible information, making it even harder to extract useful data.
> If only a single server is corrupted, this does not compromise individual user privacy. For instance, with server $S_0$, as long as the underlying encryption algorithm is secure, the server cannot access the user-submitted private data without the user's private key. Similarly, for server $S_1$, the security of the underlying Separable Homomorphic Commitment (SHC) ensures that its hiding properties prevent $S_1$ from obtaining any private information.
> In conclusion, the assumptions of our scheme are reasonable and well-supported. We will incorporate these clarifications in the revised version to better highlight the theoretical advantages of our approach.
>
> 2. $\textbf{Adding Experiments}$. Our scheme is not limited to specific models or datasets. In the paper, we have conducted theoretical analysis and experimental comparisons of each scheme's performance under various models and datasets (lines 362-564). Theoretical analysis indicates that our scheme significantly improves computational efficiency. However, to better showcase the advantages of our scheme, we will include additional experiments in future versions to verify performance from the perspectives of computational and communication costs. Furthermore, to address your concerns, we will add more datasets, such as CIFAR-100 and Fashion-MNIST, and include new SOTA comparison schemes like Elsa (SP’23) and VeriFL (IEEE TIFS’20) in the updated version.
>
> 3. $ \textbf{Comparison with Other Schemes}$. We have already included several representative advanced schemes of the same type in the paper, but we will still add Elsa (SP’23) and VeriFL (IEEE TIFS’20) as comparison schemes. Compared to Elsa (SP’23), our scheme makes weaker assumptions, resulting in higher security while supporting multi-round aggregation with a significant performance improvement. Compared to VeriFL (IEEE TIFS’20), our scheme does not require constructing complex communication graphs or performing time-consuming secret sharing operations, which leads to substantial performance gains.
>
> 4. $\textbf{Additional Experiments}$. Thank you for your suggestion. In the paper, we have already provided single-round timing statistics for both the client and server across different models and datasets. The experiment you mentioned is feasible to add, and we will include these updated metrics in the revised version.
> We greatly appreciate your feedback and will ensure these clarifications are incorporated in the revised paper. Thank you once again, and we look forward to your response.

---

> > ### Author Response · Authors · 2024-11-20
> > **Revised paper uploaded**
> >
> > Thank you once again for your time and valuable feedback on our work. We have thoroughly addressed your concerns in the revised paper, which has now been submitted. In response to your concerns, we have made the following changes.
> >
> > $\textbf{Uncolluding Assumptions:}$ We have revised the uncolluding assumptions, which are now included in Section C.1 for clarity and comprehensiveness.
> >
> > $\textbf{More SOTA Comparison Schemes:} $ We have added two new SOTA schemes in Section D.1 to analyze and compare them theoretically.
> >
> > $\textbf{Additional Experiments:}$ Our scheme is not limited to specific models or datasets. To better support this conclusion, we have added more SOTA methods and experiments as your suggestions.
> > 1. Sections D.1 and D.2 now incorporate two new SOTA methods, VeriFL and ELSA. We provide both theoretical and experimental analyses to highlight their implications and comparisons.
> > 2. In Section D.2, we also add more experimental results using the more complex CIFAR-100 dataset across different models to further validate our conclusions.
> >
> > We believe these revisions address your concerns and enhance the paper. Please let us know if you have any further questions, and we look forward to your feedback.

---

> > > ### Comment · Reviewer_3v49 · 2024-12-03
> > >
> > > Thanks to the author for replying. The author has solved my problem in the new version of the manuscript, so I maintain my original score.

---

### Official Review · Reviewer_JhGp · 2024-11-05

**Soundness:** 3
**Presentation:** 2
**Contribution:** 3
**Rating:** 6
**Confidence:** 3

**Summary:**

This paper consider several key challenges in secure aggregation: dynamic user paticipation, resistance to model inconsistency attacks (MIA), and verifiability of aggregation of malcious servers. This paper proposes a dual-server architecture where one server aggregates the masked gradients and the other aggregates the masks, ensuring that neither server has access to the final aggregation result, thus protecting against MIA. It also incorporates a novel cryptographic primitive, Separable Homomorphic Commitment (SHC), which enables clients to verify the correctness of the server’s aggregation without sacrificing efficiency.

**Strengths:**

-  While a two-server model is not new, I like the idea that the proposed method introduces the dual server model to protect against MIA by preventing either server from accessing the final aggregation result.
- This paper introduces SHC, which allows users to verify aggregation correctness without incurring heavy computational costs.
- It reduces the communication and computation overhead from logarithmic to constant scale, which is a major improvement over advanced schemes like Flamingo and BBSA. This makes it more practical for large-scale federated learning frameworks.

**Weaknesses:**

- The proposed method relies heavily on the assumption that the two servers do not collude. While this assumption is reasonable in certain applications, it is also a potential limitation. In practice, ensuring non-collusion between two entities may not always be feasible, especially in untrusted environments.
- In addition, while the paper claims that the proposed scheme mitigates the risks associated with a single-server setup, the system still relies on the assumption that both servers should be successful in aggregation. If either server fails, the entire system could be at risk.
- As the SHC protocol plays key role to verify the correctness in the dual-server system, it would be helpful if the SHC protocol is described with clearer notation and more intuitive explanations. For instance, separation of commitments could be explained more thoroughly for readers unfamiliar with the cryptographic concepts.

**Questions:**

please see the comments in weakness.

---

> ### Author Response · Authors · 2024-11-14
>
> We sincerely thank you for your time and valuable feedback. Below, we address the technical and theoretical concerns you raised.
> 1) $\textbf{Uncolluding Assumption}$. The assumption of two uncolluding servers is common in federated learning research, as demonstrated by works like Elsa (SP’23), VeriFL (IEEE TIFS’20), and Flamingo (SP’23), all of which make similar assumptions. In our scheme, while we assume the servers are uncolluding, they are allowed to behave maliciously during aggregation. This design choice is supported by our scheme’s verifiability mechanism, which ensures that any malicious aggregation results can be detected. Furthermore, our scheme allows for up to $n-2$ colluding clients, which is a reasonable and sufficient level of security, even in environments where trust is limited.
> 2) $\textbf{Server Aggregation Verification}$. As outlined in our previous response, our scheme includes a verification mechanism to ensure the correctness of the server’s aggregation results. If an incorrect aggregation occurs, our system can detect the anomaly through this verification. To further incentivize accurate aggregation by the servers, future research could introduce mechanisms such as reputation scores, rewarding servers that perform correct aggregation. Additionally, we will revise the paper to correct the assumption of semi-honest servers, clarifying that servers can act maliciously, which is fully accounted for in our design.
> 3) $\textbf{SHC Instantiation and Explanation}$. To aid reader understanding, we provide an instantiation of Separable Homomorphic Commitments (SHC) in lines 245–289. We also offer a more detailed description of the SHC instantiation in lines 803–863. To improve the clarity of our explanation, we will revise the paper to include a more intuitive explanation of SHC and the other cryptographic concepts used in our scheme.
>
> We greatly appreciate your feedback and will incorporate these clarifications in the revised version of the paper. Thank you again, and we look forward to your response.

---

> > ### Author Response · Authors · 2024-11-20
> > **Revised paper uploaded**
> >
> > Thanks again for your time and valuable feedback on our work. We have carefully addressed your concerns in the revised paper, which has just been uploaded.
> >
> > In response to your comments, we have revised the uncolluding assumptions, which are now included in Section C.1 for clarity and comprehensiveness. Specifically, we have revised the paper to correct the assumption of semi-honest servers, clarifying that servers can act maliciously, which is fully accounted for in our design in Section C.1.
> >
> > We believe that these revisions thoroughly address your concerns and strengthen the paper. If you have any further questions or concerns, please do not hesitate to reach out. We look forward to your feedback.

---

### Author Response · Authors · 2024-11-26
**General Response**

Dear Reviewers and  Area Chair,

Thanks to all reviewers and AC for your efforts and time. We hope these revisions address your concerns and improve the clarity and quality of the paper. However, a few days have passed since we submitted our reply and revision file. We wanted to check if our responses and updates adequately address your concerns. If you have any additional questions or comments regarding our work, we would be glad to hear from you. Once again, thank you for your valuable feedback and support.

Best regards

---

### Meta-Review · Area_Chair_yX4s · 2024-12-08

**Metareview:**

The paper presents a multi-round secure aggregation (SA) scheme for federated learning.
Reviewers noted that some of the assumptions in this paper are unrealistic.
Besides, there are other unresolved issues such as unclear security proof and Incomplete comparison.
The authors' rebuttal did not sufficiently address these issues, and the reviewers have maintained their scores.
Given these issues, I recommend rejection.

**Additional Comments On Reviewer Discussion:**

Both Reviewer sXBs and Reviewer Xv5P raised concerns about the threat model, security proof, and the need for more detailed experiments. The authors' rebuttal did not sufficiently address these issues, and the reviewers have maintained their scores. Given the two rejection recommendations, I believe the paper does not meet the threshold for acceptance.

---

### Decision · Program_Chairs · 2025-01-22

Reject